# Foundation neural-networks quantum states as a unified Ansatz for multiple hamiltonians

Riccardo Rende [1,6] ✉, Luciano Loris Viteritti[2,6] ✉, Federico Becca[3], Antonello Scardicchio[4,5], Alessandro Laio [1,4] & Giuseppe Carleo [2] ✉

Foundation models are highly versatile neural-network architectures capable of processing different data types, such as text and images, and generalizing across various tasks like classification and generation. Inspired by this success, we propose Foundation Neural-Network Quantum States (FNQS) as an integrated paradigm for studying quantum many-body systems. FNQS leverage key principles of foundation models to define variational wave functions based on a single, versatile architecture that processes multimodal inputs, including spin configurations and Hamiltonian physical couplings. Unlike specialized architectures tailored for individual Hamiltonians, FNQS can generalize to physical Hamiltonians beyond those encountered during training, offering a unified framework adaptable to various quantum systems and tasks. FNQS enable the efficient estimation of quantities that are traditionally challenging or computationally intensive to calculate using conventional methods, particularly disorder-averaged observables. Furthermore, the fidelity susceptibility can be easily obtained to uncover quantum phase transitions without prior knowledge of order parameters. These pretrained models can be efficiently fine-tuned for specific quantum systems. The architectures trained in this paper are publicly available at https://huggingface.co/nqs-models, along with examples for implementing these neural networks in NetKet.

The field of machine learning has undergone a fundamental transformation with the emergence of foundation models[1]. Built upon the Transformer architecture[2], these models have transcended their origins in language tasks[3,4] to establish new paradigms across domains, from image generation[5] to protein structure prediction[6,7]. Their efficacy emerges from a profound empirical observation: the scaling of models to hundreds of billions of parameters enables task-agnostic learning that achieves parity with specialized approaches while generating solutions for arbitrary problems defined at inference time[8]. These models exhibit remarkable generalization capabilities, enabling them to adapt to an extensive variety of tasks and domains without requiring task-specific fine-tuning. Another essential feature is their multimodality: they are trained on datasets comprising various formats, including text, images, videos, and audio, allowing them to process and generate outputs that combine these different forms. Foundation models have led to an unprecedented level of homogenization: almost all state-of-the-art natural language processing models are now adapted from a few foundation models. This homogenization produces extremely high leverage since enhancements to foundation models can directly and broadly improve performance across various applications.

In parallel, the study of quantum many-body systems has been significantly impacted by neural-network architectures employed as variational wave functions[9]. Neural-Network Quantum States (NQS) have emerged as a powerful framework for describing strongly-correlated models with unprecedented accuracy[10–14]. Recent advances

[1]International School for Advanced Studies (SISSA), Trieste, Italy. [2]Institute of Physics, École Polytechnique Fédérale de Lausanne (EPFL), Lausanne, Switzerland. [3]Dipartimento di Fisica, Università di Trieste, Trieste, Italy. [4]The Abdus Salam ICTP, Trieste, Italy. [5]INFN, Sezione di Trieste, Trieste, Italy. [6]These authors contributed equally: Riccardo Rende, Luciano Loris Viteritti. ✉e-mail: rrende@sissa.it; luciano.viteritti@epfl.ch; giuseppe.carleo@epfl.ch

in Stochastic Reconfiguration[15–17] have enabled the stable optimization of variational states with millions of parameters[18,19], while the adaptation of the Transformer architecture for NQS parametrization[20–25] has achieved state-of-the-art performance in challenging systems[19,21]. Despite this progress, NQS are typically conceived in a system-specific fashion, and studying different Hamiltonians requires significant efforts both in design and numerical optimization strategies.

To address these limitations, we present here Foundation Neural-Network Quantum States (FNQS), a theoretical framework that synthesizes these advances by training neural-network-based variational wave functions capable of integrating as input not only the "standard" basis on which the wave function is represented, but also detailed information about the Hamiltonian (see Fig. 1). Our architecture is designed to achieve three key characteristics of foundation models in the quantum context: multimodality, through the ability to process multiple input types such as spin configurations and physical couplings; homogenization, by applying a single architecture across different Hamiltonians from simple to disordered systems; and generalization to physical Hamiltonians beyond the training dataset.

Previous efforts to construct foundation model-inspired wave functions have been reported in refs. 26–30. However, these approaches exhibit several limitations that are addressed in the present work. Specifically, some studies have been constrained to simple physical systems, achieving limited accuracy compared to specialized approaches[26], while others have employed ad hoc optimization strategies for chemical systems[27–29].

In contrast, our work demonstrates applications that are unprecedented in both the diversity and complexity of physical models tackled by a single foundation model. We systematically explore systems of increasing complexity, including two-dimensional frustrated magnets with multiple couplings and disordered systems. This is enabled by the introduction of a suitably designed neural-network wave function based on the Transformer architecture[2,5], combined with an optimization strategy that extends the Stochastic Reconfiguration method[15,16] to simultaneously optimize across multiple systems. This generalized optimization procedure is essential to achieving accurate results in the variational Monte Carlo framework.

Most notably, our framework enables simultaneous optimization of wave functions for multiple systems with computational complexity equivalent to single-system optimization, with no performance degradation as the number of systems increases. In addition, the framework enables efficient estimation of the fidelity susceptibility[31] (see Methods), providing rigorous, unsupervised detection of quantum phase transitions without prior knowledge of the order parameters[32,33]. Refer to Fig. 1 for a pictorial representation of the different applications.

In this work, we develop the theoretical framework for simultaneous training of variational wave functions across multiple quantum systems, adapting both Stochastic Reconfiguration for multi-system optimization and the Transformer architecture for multimodal quantum state parametrization. We present systematic validation on the exactly solvable transverse field Ising model in one dimension, followed by an investigation of the $J_1$-$J_2$-$J_3$ Heisenberg model on a square lattice through fidelity susceptibility analysis. We conclude with an examination of disordered Hamiltonians, demonstrating the framework's capacity for efficient estimation of disorder-averaged quantities.

## Results
### Theoretical framework

The first step in developing foundation models to approximate ground states of quantum many-body Hamiltonians is to establish a theoretical framework that enables training a single NQS to approximate the ground states of multiple systems simultaneously. Consider a family of Hamiltonians, denoted by $\hat{H}_{\boldsymbol{\gamma}}$, where $\boldsymbol{\gamma}$ is a set of parameters that characterize each specific Hamiltonian, such as the physical couplings. Our goal is to find an approximation of the ground state of the ensemble of Hamiltonians $\hat{H}_{\boldsymbol{\gamma}}$ using a variational wave function $|\psi_\theta(\boldsymbol{\gamma})\rangle$ which explicitly depends on the physical couplings $\boldsymbol{\gamma}$ and on a shared set of variational parameters $\theta$ for all the Hamiltonians. To this end, we define the following loss function:

$$\mathcal{L}(\theta) = \int d\boldsymbol{\gamma}\, \mathcal{P}(\boldsymbol{\gamma}) \frac{\langle \psi_\theta(\boldsymbol{\gamma})|\hat{H}_{\boldsymbol{\gamma}}|\psi_\theta(\boldsymbol{\gamma})\rangle}{\langle \psi_\theta(\boldsymbol{\gamma})|\psi_\theta(\boldsymbol{\gamma})\rangle}, \tag{1}$$

where $\mathcal{P}(\boldsymbol{\gamma})$ is a normalized probability density over the couplings, i.e., $\int d\boldsymbol{\gamma}\mathcal{P}(\boldsymbol{\gamma}) = 1$. We denote expectation values with respect to the variational state $|\psi_\theta(\boldsymbol{\gamma})\rangle$ as $\langle \cdots \rangle_{\boldsymbol{\gamma}}$. This loss function represents an ensemble average of the energy expectation value $\langle \hat{H}_{\boldsymbol{\gamma}} \rangle_{\boldsymbol{\gamma}}$, weighted by the distribution $\mathcal{P}(\boldsymbol{\gamma})$. For each value of $\boldsymbol{\gamma}$, the variational energy $\langle \hat{H}_{\boldsymbol{\gamma}} \rangle_{\boldsymbol{\gamma}}$ is bounded from below by the exact ground state energy $E_0(\boldsymbol{\gamma})$, such that $\langle \hat{H}_{\boldsymbol{\gamma}} \rangle_{\boldsymbol{\gamma}} \geq E_0(\boldsymbol{\gamma})$. Consequently, the loss function in Eq. (1) is bounded as

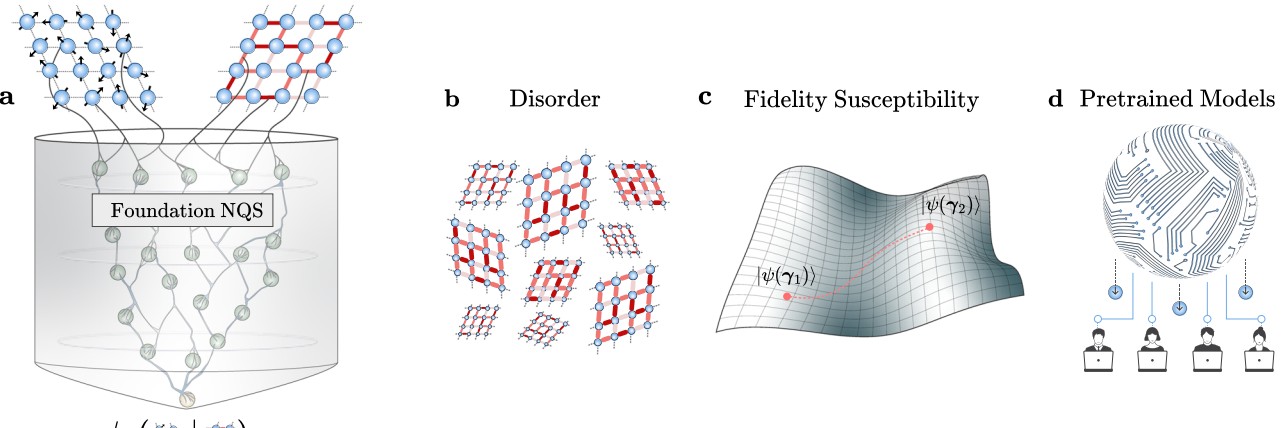

**a**                **b**  Disorder          **c**  Fidelity Susceptibility          **d**  Pretrained Models

Foundation NQS

$\psi_\theta\big(\,\big|\,\big)$

**Fig. 1 | Pictorial representation and applications of Foundation Neural-Network Quantum States.** The panel (**a**) shows a pictorial representation of Foundation Neural-Network Quantum States (FNQS), which, unlike traditional NQS, process multimodal inputs by incorporating both physical configurations and Hamiltonian couplings to define a variational wave function amplitude over their joint space. FNQS enable a range of applications, including the efficient simulation of disordered systems [see panel (**b**)] and the estimation of the quantum geometric tensor in coupling space, also known as the fidelity susceptibility, for the unsupervised detection of quantum phase transitions [see panel (**c**)]. Moreover, FNQS combined with the public availability of the architectures allows users to leverage pretrained models to explore coupling regimes beyond those encountered during training [see panel (**d**)].

$\mathcal{L}(\theta) \geq \mathcal{L}_0$, where $\mathcal{L}_0 = \int d\boldsymbol{\gamma} \mathcal{P}(\boldsymbol{\gamma}) E_0(\boldsymbol{\gamma})$ is the average ground state energy over the distribution $\mathcal{P}(\boldsymbol{\gamma})$.

The loss function in Eq. (1) can equivalently be written in a form amenable for Monte Carlo averages:

$$\mathcal{L}(\theta) = \int d\boldsymbol{\gamma}\, \mathcal{P}(\boldsymbol{\gamma}) \sum_{\boldsymbol{\sigma}} \frac{|\psi_\theta(\boldsymbol{\sigma}|\boldsymbol{\gamma})|^2}{\langle \psi_\theta(\boldsymbol{\gamma})|\psi_\theta(\boldsymbol{\gamma})\rangle} E_L(\boldsymbol{\sigma},\boldsymbol{\gamma}). \qquad (2)$$

Here, we have introduced the local energy $E_L(\boldsymbol{\sigma},\boldsymbol{\gamma}) = \langle \boldsymbol{\sigma}|\hat{H}_{\boldsymbol{\gamma}}|\psi_\theta(\boldsymbol{\gamma})\rangle / \langle \boldsymbol{\sigma}|\psi_\theta(\boldsymbol{\gamma})\rangle$ and the wave function $\langle \boldsymbol{\sigma}|\psi_\theta(\boldsymbol{\gamma})\rangle = \psi_\theta(\boldsymbol{\sigma}|\boldsymbol{\gamma})$. The latter is parametrized by a neural network and is the core variational object in our framework. Importantly, the explicit dependence of the many-body wave function amplitude $\psi_\theta(\boldsymbol{\sigma}|\boldsymbol{\gamma})$ on the Hamiltonian couplings $\boldsymbol{\gamma}$ is a major difference compared to traditional NQS and aligns with the principles of foundation models, where the capability to handle multiple data modalities, commonly referred to as *multimodality*, plays a central role (see Fig. 1). The expectation value of any generic operator which is written in the form of Eq. (1) can be stochastically estimated using the Variational Monte Carlo framework[17], as discussed in Methods. In what follows, we denote by $M$ the number of physical configurations used for the stochastic estimation of observables across $\mathcal{R}$ systems. Assuming that the samples are equally distributed across the systems, the number of samples per system is $M/\mathcal{R}$.

The structure of the probability distribution $\mathcal{P}(\boldsymbol{\gamma})$ depends on the specific application. In disordered systems, a set of couplings $\{\boldsymbol{\gamma}_1, \ldots, \boldsymbol{\gamma}_{\mathcal{R}}\}$ can be directly sampled from $\mathcal{P}(\boldsymbol{\gamma})$, which may have continuous or discrete support. Conversely, in non-disordered systems, the probability distribution can be defined as $\mathcal{P}(\boldsymbol{\gamma}) = 1/\mathcal{R} \sum_{k=1}^{\mathcal{R}} \delta(\boldsymbol{\gamma} - \boldsymbol{\gamma}_k)$, where $\boldsymbol{\gamma}_k$ denotes the specific instances of the $\mathcal{R}$ Hamiltonians under study.

## Foundation neural-network architecture

To parametrize the FNQS, we adapt the Vision Transformer (ViT) Ansatz introduced in ref. 21 to process multimodal inputs, defined by the physical configurations $\boldsymbol{\sigma}$ and the Hamiltonian couplings $\boldsymbol{\gamma}$.

The traditional ViT architecture processes the physical configuration $\boldsymbol{\sigma}$ in three main steps (see ref. 21 for a detailed description):

1. *Embedding*. The input configuration $\boldsymbol{\sigma}$ is split into $n$ patches, where the specific shape of the patches depends on the structure of the lattice and its dimensionality, see for example refs. 20,21,23. Then, the patches are embedded in $\mathbb{R}^d$ through a linear transformation of trainable parameters, defining a sequence of input vectors $(\boldsymbol{x}_1, \boldsymbol{x}_2, \ldots, \boldsymbol{x}_n)$.
2. *Transformer Encoder*. The resulting input sequence is processed by a Transformer Encoder, which produces another sequence of vectors $(\boldsymbol{y}_1, \boldsymbol{y}_2, \ldots, \boldsymbol{y}_n)$, with $\boldsymbol{y}_i \in \mathbb{R}^d$ for all $i$.
3. *Output layer*. These vectors are summed to produce the hidden representation $\boldsymbol{z} = \sum_{i=1}^{n} \boldsymbol{y}_i$, which is finally mapped through a fully-connected layer to a single complex number representing the amplitude corresponding to the input configuration. Only the parameters of this last layer are taken to be complex-valued.

The generalization of the architecture to include as inputs the couplings $\boldsymbol{\gamma}$ is performed by modifying *only* the *Embedding* step described above. In particular, we adopt two different strategies, which cover the systems studied in this work, depending on whether the parameter vector $\boldsymbol{\gamma}$ consists of $O(1)$ or $O(N)$ real numbers, with $N$ indicating the total number of physical degrees of freedom of the model. We stress that the property of having a single, versatile architecture that can be adapted to study physical systems with distinct characteristics, such as a different number of couplings, is a key property of foundation models, also called *homogenization*. In the first scenario where the auxiliary parameters are $O(1)$, we concatenate the

values of the couplings to each patch of the physical configuration before the linear embedding. Then the usual linear embedding procedure in $\mathbb{R}^d$ is performed. Instead, in the second scenario with $O(N)$ external parameters, we split the vector of the couplings into patches using the same criterion used for the physical configuration. We then use two different embedding matrices to embed the resulting patches of the configuration and of the couplings, generating two sequences of vectors: $(\boldsymbol{x}_1, \boldsymbol{x}_2, \ldots, \boldsymbol{x}_n)$ with $\boldsymbol{x}_i \in \mathbb{R}^{d/2}$ for the physical degrees of freedom and $(\tilde{\boldsymbol{x}}_1, \tilde{\boldsymbol{x}}_2, \ldots, \tilde{\boldsymbol{x}}_n)$ with $\tilde{\boldsymbol{x}}_i \in \mathbb{R}^{d/2}$ for the couplings. The final input to the Transformer is constructed by concatenating the embedding vectors, forming the sequence $(\text{Concat}(\boldsymbol{x}_1, \tilde{\boldsymbol{x}}_1), \ldots, \text{Concat}(\boldsymbol{x}_n, \tilde{\boldsymbol{x}}_n))$, with $\text{Concat}(\boldsymbol{x}_i, \tilde{\boldsymbol{x}}_i) \in \mathbb{R}^d$. Notice that after the first layer, the representations of the configurations and of the couplings are mixed by the attention mechanism. The Embedding step can be generalized to any general parameterized Hamiltonian represented as a graph[34].

Regarding the lattice symmetries encoded in the architecture, for non-disordered Hamiltonians we employ a translationally invariant attention mechanism that ensures a variational state invariant under translations among patches[21,23]. In contrast, for disordered models, we do not impose constraints on the attention mechanism.

## Transverse field Ising chain

In the first place, we test the framework on the one-dimensional Ising model in a transverse field, an established benchmark problem of the field. The system is described by the following Hamiltonian (with periodic boundary conditions):

$$\hat{H} = -J \sum_{i=1}^{N} \hat{S}_i^z \hat{S}_{i+1}^z - h \sum_{i=1}^{N} \hat{S}_i^x, \qquad (3)$$

where $\hat{S}_i^x$ and $\hat{S}_i^z$ are spin-1/2 operators on site $i$. The ground-state wave function, for $J, h \geq 0$, is positive definite in the computational basis, with a known exact solution. In this case, the Hamiltonian depends on a single coupling, specifically the ratio $h/J$.

In the thermodynamic limit, the ground state exhibits a second-order phase transition at $h/J = 1$, from a ferromagnetic ($h/J < 1$) to a paramagnetic ($h/J > 1$) phase. In finite systems with $N$ sites, the estimation of the critical point can be obtained from the long-range behavior of the spin-spin correlations, that is, $m^2(\boldsymbol{\gamma}) = 1/N \sum_{i=1}^{N} \langle \hat{S}_i^z \hat{S}_{i+N/2}^z \rangle_{\boldsymbol{\gamma}}$. The quantum phase transition at $h/J = 1$ is in the universality class of the classical two-dimensional Ising model[35].

Here, we first demonstrate the ability to train a FNQS across multiple Hamiltonians, and even across quantum phase transitions. To achieve this, we train a FNQS on a chain of $N = 100$ sites across five different values of the external field ($\mathcal{R} = 5$), including values representative of both the disordered ($h/J = 1.2, 1.1$) and the magnetically ordered phase ($h/J = 0.9, 0.8$), as well as the transition point ($h/J = 1.0$). As shown in Fig. 2a, this single neural network describes all five ground states with high accuracy. The learning speed is only moderately different in the different states. In particular, the state with a value of $h/J$ close to the transition point is the one that converges last. For the same architecture, we systematically vary the value of $\mathcal{R} \in [5, 2000]$, choosing the transverse field equispaced within the interval $h/J \in [0.8, 1.2]$. We keep the total batch size fixed to $M = 10000$, assigning an equal number of samples $M/\mathcal{R}$ across the $\mathcal{R}$ different systems. In the inset of panel (a), we show the relative error of the total energy accuracy as a function of $\mathcal{R}$. Remarkably, despite the number of systems increasing, the network's performance remains constant, with no observable degradation in accuracy. Crucially, this robustness is achieved at a *computational cost independent of the total number of systems*, as it depends solely on the neural network architecture and the fixed total batch size $M$. This result is a first illustration of the accuracy, scalability, and computational efficiency of our approach.

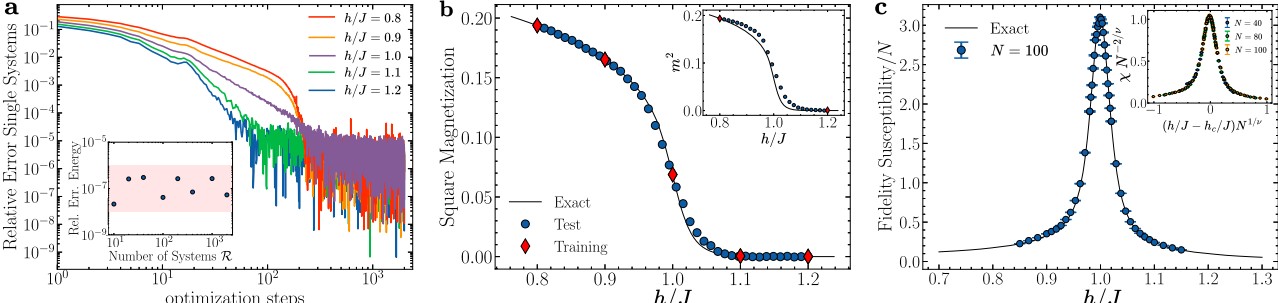

**Fig. 2 | Transverse field Ising on a chain. a** Simultaneous ground state energy optimization of $\mathcal{R} = 5$ systems on a chain of $N = 100$ sites, with $h/J = 0.8, 0.9, 1.0, 1.1$ and 1.2. The relative error with respect to the exact ground state energy of each system is shown as a function of the optimization steps. The inset displays the relative error of the total energy as a function of the number of systems $\mathcal{R}$, defined by equispaced values of $h/J$ in the interval $h/J \in [0.8, 1.2]$, with a fixed batch size of $M = 10000$. **b** Square magnetization evaluated with a FNQS trained at $h/J = 0.8, 0.9, 1.0, 1.1$ and 1.2 (red diamonds) and tested on previously *unseen* values of the external field (blue circles). The inset shows the square magnetization predictions of an architecture trained exclusively on $h/J = 0.8$ and 1.2, evaluated at intermediate external field values. **c** Fidelity susceptibility per site [see Eq. (22)] as a function of the external field for a FNQS trained on $\mathcal{R} = 6000$ equispaced values of $h/J$ in the interval $h/J \in [0.85, 1.15]$ for a cluster of $N = 100$ sites. The inset shows the data collapse of the same quantity for $N = 40, 80$, and 100.

Then, we investigate the generalization properties of the FNQS. In panel (b) of Fig. 2, we use the architecture trained with $\mathcal{R} = 5$ and evaluate its performance on external field values not included in the training set. In particular, we compute the square magnetization for other intermediate values of $h/J$, showing robust generalization capabilities of the network across the entire phase diagram. The inset of the same plot explores a more restricted scenario in which training is performed using only two points: one in the disordered phase ($h/J = 1.2$) and another in the ordered phase ($h/J = 0.8$). This analysis shows that, even with minimal training data, the network avoids overfitting the ground state at these two points and learns a sufficiently smooth description of the magnetization curve.

Finally, in panel (c) of Fig. 2, we use a FNQS trained on $\mathcal{R} = 6000$ different points equispaced in the interval $h/J \in [0.85, 1.15]$ to calculate the fidelity susceptibility $\chi(\boldsymbol{\gamma})$ [see Eq. (21) in Methods], comparing the FNQS results to the exact solution that is available in this case[36,37]. In the inset of the same panel, we present a data collapse analysis of the fidelity susceptibility. Specifically, we show the scaled fidelity susceptibility $\chi N^{-2/\nu}$ versus $(h/J - h_c/J)N^{1/\nu}$ according to the scaling laws of refs. 31,32,38,39. The data collapses well under $h_c/J = 1.00(1)$ and the critical exponent $\nu = 1.00(2)$ corresponding to the classical two-dimensional Ising universality class[40].

This first benchmark example highlights the ability of the FNQS to interpolate meaningfully between different phases, even when trained on a limited set of Hamiltonians. We attribute this capability to the properties of the ViT architecture employed. In particular, the multi-head attention mechanism could play a crucial role. For example, each attention head can, in principle, specialize in capturing features associated with distinct phases of the system. Moreover, the all-to-all connectivity intrinsic to the attention mechanism allows the network to flexibly describe long-range correlations, which are essential for accurately describing critical phenomena.

### $J_1$-$J_2$-$J_3$ Heisenberg model

We now proceed to analyzing the $J_1$-$J_2$-$J_3$ Heisenberg model on a two-dimensional $L \times L$ square lattice with periodic boundary conditions:

$$\hat{H} = J_1 \sum_{\langle \boldsymbol{r}, \boldsymbol{r}' \rangle} \hat{\boldsymbol{S}}_{\boldsymbol{r}} \cdot \hat{\boldsymbol{S}}_{\boldsymbol{r}'} + J_2 \sum_{\langle\langle \boldsymbol{r}, \boldsymbol{r}' \rangle\rangle} \hat{\boldsymbol{S}}_{\boldsymbol{r}} \cdot \hat{\boldsymbol{S}}_{\boldsymbol{r}'} + J_3 \sum_{\langle\langle\langle \boldsymbol{r}, \boldsymbol{r}' \rangle\rangle\rangle} \hat{\boldsymbol{S}}_{\boldsymbol{r}} \cdot \hat{\boldsymbol{S}}_{\boldsymbol{r}'}, \quad (4)$$

where $\hat{\boldsymbol{S}}_{\boldsymbol{r}} = (\hat{S}_{\boldsymbol{r}}^x, \hat{S}_{\boldsymbol{r}}^y, \hat{S}_{\boldsymbol{r}}^z)$ represents the spin-1/2 operator localized at site $\boldsymbol{r}$; in addition, $J_1$, $J_2$, and $J_3$ are first-nearest-, second-nearest-, and third-nearest-neighbor antiferromagnetic couplings, respectively. The ground-state properties of this frustrated model have been extensively

studied using various numerical and analytical approaches. However, a complete characterization of its phase diagram remains challenging[41–48]. It is well established that antiferromagnetic order dominates in extended regions for $J_1 \gg J_2, J_3$ [with pitch vector $\boldsymbol{k} = (\pi, \pi)$] and for $J_2 \gg J_1, J_3$ [with pitch vectors $\boldsymbol{k} = (\pi, 0)$ or $\boldsymbol{k} = (0, \pi)$]. In contrast, in the intermediate region, frustration suppresses magnetic order, leading to valence-bond solid and, as recently suggested, spin-liquid states[47,48]. The study of this model using FNQS aims to demonstrate that a single architecture can learn to effectively combine input spin configurations and Hamiltonian couplings, constructing a compact representation that captures and differentiates between distinct phases.

First, we aim for an initial characterization of the phase diagram in a fully unsupervised manner, aiming to distinguish regions with valence-bond ground states from those with magnetic order using the generalized fidelity susceptibility (see Methods). To this end, we train a FNQS on a $10 \times 10$ lattice over a broad region of parameter space, setting a dense grid of $\mathcal{R} = 4000$ evenly spaced points in the plane defined by $J_2/J_1 \in [0, 1.0]$ and $J_3/J_1 \in [0, 0.6]$. Having two couplings $J_2/J_1$ and $J_3/J_1$, the quantum geometric tensor in the couplings space $\chi(\boldsymbol{\gamma})$ [see Eq. (22) of Methods] is a $2 \times 2$ matrix. For each point $\boldsymbol{\gamma} = (J_2/J_1, J_3/J_1)$ we diagonalize $\chi(\boldsymbol{\gamma})$ and in Fig. 3a we visualize the direction of the eigenvector corresponding to the maximum eigenvalue using lines, whose colors are associated with the leading eigenvalues and indicate the intensity of maximum variation of the variational wave function. We note that the lines of maximal variation partition the plane into three distinct regions, in agreement with the three different phases identified by the order parameters (see below). Remarkably, within this approach we are able to identify the existence of two phase transitions without any prior knowledge of the physical properties of the system. Furthermore, by analyzing the behavior of the eigenvectors, we can infer the nature of these phase transitions. For example, on the left branch of maximum variation, the eigenvectors exhibit no significant change in direction before and after the transition, which is indicative of a continuous phase transition. In contrast, the right branch shows a pronounced change in the eigenvector directions across the transition, suggesting a first-order phase transition. To the best of our knowledge, this is the first calculation of fidelity susceptibility for a system with more than one coupling. Indeed, without our approach, it would be highly computationally expensive to optimize thousands of systems with different coupling values, using finite difference methods to estimate the geometric tensor in the couplings space [see Eq. (22) in Methods].

To further analyze the physical property of the model, we compute the order parameters in each region of the phase diagram by

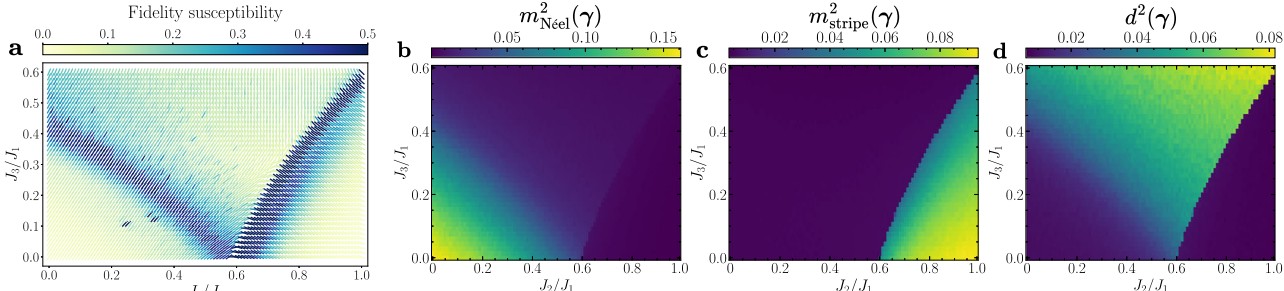

**Fig. 3 | Fidelity susceptibility and order parameters of the $J_1$-$J_2$-$J_3$ Heisenberg model. a** Fidelity susceptibility of the $J_1$-$J_2$-$J_3$ Heisenberg model on a $10 \times 10$ square lattice [see Eq. (4)]. For each point of the phase diagram of the system, we visualize the direction of the leading eigenvector of the quantum geometric tensor $\chi(\gamma)$ [see Eq. (22)]. The colour associated to each line is related to corresponding eigenvalue clipped in the interval [0.0, 0.5] for visualization purposes. **b** The order parameter $m^2_{\text{Néel}}(\gamma)$ characterizing the Néel antiferromagnetic order. **c** The order parameter $m^2_{\text{stripe}}(\gamma)$ identifying the antiferromagnetic phase with stripe order. **d** The order parameter $d^2(\gamma)$ probing the valence bond phase. In all panels, the order parameters are computed over a dense grid of $\mathcal{R} = 4000$ uniformly distributed points in the parameter space defined by $J_2/J_1 \in [0, 1.0]$ and $J_3/J_1 \in [0, 0.6]$.

examining spin-spin and dimer-dimer correlations. Specifically, for fixed values of the Hamiltonian couplings $\gamma = (J_2/J_1, J_3/J_1)$, the anti-ferromagnetic orders are detected by analyzing the spin structure factor

$$C(\mathbf{k}; \gamma) = \sum_{\mathbf{r}} e^{i\mathbf{k}\cdot\mathbf{r}} \left\langle \hat{\mathbf{S}}_{\mathbf{0}} \cdot \hat{\mathbf{S}}_{\mathbf{r}} \right\rangle_{\gamma}, \qquad (5)$$

where $\mathbf{r}$ runs over all the lattice sites of the square lattice. On the one side, the antiferromagnetic Néel order is detected by measuring $m^2_{\text{Néel}}(\gamma) = C(\pi, \pi; \gamma)/N$[49,50] with $N = L^2$. On the other side, the stripe antiferromagnetic order is identified by $m^2_{\text{stripe}}(\gamma) = [C(0, \pi; \gamma) + C(\pi, 0; \gamma)]/(2N)$. Furthermore, the valence-bond solid order is detected by the dimer-dimer correlations:

$$D_\alpha(\mathbf{r}; \gamma) = 9\left[ \left\langle \hat{S}^z_{\mathbf{0}} \hat{S}^z_{\mathbf{\alpha}} \hat{S}^z_{\mathbf{r}} \hat{S}^z_{\mathbf{r}+\mathbf{\alpha}} \right\rangle_{\gamma} - \left\langle \hat{S}^z_{\mathbf{0}} \hat{S}^z_{\mathbf{\alpha}} \right\rangle_{\gamma} \left\langle \hat{S}^z_{\mathbf{r}} \hat{S}^z_{\mathbf{r}+\mathbf{\alpha}} \right\rangle_{\gamma} \right], \qquad (6)$$

where $\mathbf{\alpha} = \hat{\mathbf{x}}, \hat{\mathbf{y}}$. Notice that the previous definition involves only the $z$ component of the spin operators, which is sufficient to detect the dimer order[20,51]; however, since we consider only one component, we include a factor of 9 in Eq. (6) to account for this[52]. Then, the corresponding structure factor is expressed as $\mathcal{D}_\alpha(\mathbf{k}; \gamma) = \sum_{\mathbf{r}} e^{i\mathbf{k}\cdot\mathbf{r}} D_\alpha(\mathbf{r}; \gamma)$. The order parameter to detect the valence-bond order is defined as $d^2(\gamma) = [\mathcal{D}_x(\pi, 0; \gamma) + \mathcal{D}_y(0, \pi; \gamma)]/(2N)$.

In panels (b, c, d) of Fig. 3, we present the order parameters $m^2_{\text{Néel}}(\gamma)$, $m^2_{\text{stripe}}(\gamma)$, and $d^2(\gamma)$, which respectively characterize the antiferromagnetic Néel, antiferromagnetic stripe, and valence bond solid phases, as functions of the couplings $J_2/J_1 \in [0, 1.0]$ and $J_3/J_1 \in [0, 0.6]$. Comparing the different panels in Fig. 3, we observe a strong correspondence between the phase transition boundaries predicted by fidelity susceptibility and those identified through order parameters. This agreement validates our approach to the unsupervised detection of quantum phase transitions, even in systems with multiple couplings.

Finally, to assess the accuracy of the FNQS, we focus on the line $J_3/J_1 = 0$, allowing comparison with other techniques. In panel (a) of Fig. 4, we show the results for a $6 \times 6$ lattice, where the FNQS predictions of the order parameters $m^2_{\text{Néel}}$ and $m^2_{\text{stripe}}$ are in excellent agreement with exact diagonalization results. In panel (b) of Fig. 4, we extend this analysis to a $10 \times 10$ lattice. Since exact diagonalization is infeasible at this system size, we benchmark FNQS predictions against Quantum Monte Carlo (QMC) data at the unfrustrated point $J_2/J_1 = 0$[50] and against results from a state-of-the-art ViT architecture trained from scratch at $J_2/J_1 = 0.5$[19], demonstrating the reliability of the FNQS architecture.

## Random transverse field Ising model

A natural extension of this method involves exploring Hamiltonians with quenched disorder, by optimizing a single FNQS across distinct disorder realizations. Disordered systems are a very vast and ramified topic of research and are at the basis of a theory of complexity[53]. When quantum effects are also included, disordered systems become even more compelling, with recent works highlighting the extension of Anderson localization to a complete ergodicity breaking in interacting quantum systems[54]. These systems are notoriously resilient to numerical approaches[55] and optimizing a single FNQS across many realizations of disorder makes the averaging of the physical quantities, a necessary step for treating disordered systems, much more efficient.

A compelling candidate for study is the random transverse field Ising chain, defined by the following Hamiltonian (assuming periodic boundary conditions):

$$\hat{H} = -J \sum_{i=1}^{N} \hat{S}^z_i \hat{S}^z_{i+1} - \sum_{i=1}^{N} h_i \hat{S}^x_i, \qquad (7)$$

where $h_i$ is the on-site transverse magnetic field at the $i$-th site. In the disordered case, $h_i$ varies randomly along the chain, drawn independently and identically from the uniform distribution on the interval $[0, h_0]$. When setting $J = 1/e$, the model exhibits a quantum phase transition between ordered (ferromagnetic) and disordered (para-magnetic) phases for $h_0 = 1$[56–59]. Although this disordered model cannot be solved analytically due to the lack of translational symmetry, the eigenstates can be found efficiently for each realization of disorder by exploiting the mapping to free fermions[58]. Therefore, relatively large clusters may be considered, just requiring diagonalizations of $N \times N$ matrices[58]. This model is deceptively simple, since for a large region going from the critical point inside the disordered phase, it is affected by Griffiths-McCoy singularities[56,57].

From a numerical perspective, unlike in previous cases, the coupling distribution $\mathcal{P}(\gamma)$ is a uniform distribution for the $N$ transverse fields $h_i$ in Eq. (7). Consequently, for each realization of disorder, the number of couplings is equal to the number of sites of the lattice. This scenario provides an opportunity to assess the generalization capabilities of the neural network, particularly in its ability to accurately predict properties for new disorder realizations beyond those considered during the training.

In Fig. 5a, we optimize a single FNQS on a cluster of $N = 64$ sites. Training is carried out on $\mathcal{R}$ distinct disorder realizations, sampled by fixing $h_0 = 1$. The left (right) panel presents the relative error of the variational energy for seven different training (test) seeds as a function of the number of training realizations, namely $\mathcal{R} = 8, 20, 100, 1000$, while keeping in all cases the total batch size of spin configurations

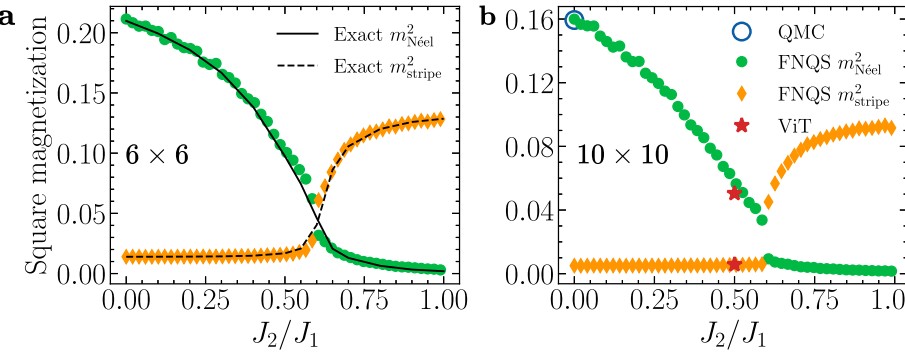

**Fig. 4 | Benchmarking FNQS on the $J_1$-$J_2$-$J_3$ Heisenberg model along the axis $J_3 = 0$. a** FNQS results for the square magnetization corresponding to the Néel ($m^2_{\text{Néel}}$) and stripe ($m^2_{\text{stripe}}$) order parameters are compared at $J_3/J_1 = 0$ with exact diagonalization calculations (solid and dashed black lines) on a $6 \times 6$ cluster.

**b** Variational estimates of the magnetic order parameters are compared with Quantum Monte Carlo (QMC, blue circles) at $J_2/J_1 = 0$, and with Vision Transformer architecture (ViT, red stars) at $J_2/J_1 = 0.5$ on a $10 \times 10$ lattice.

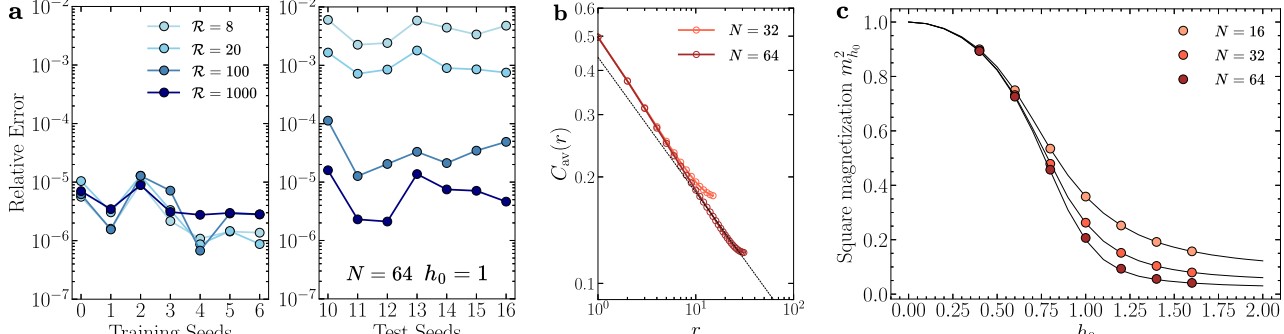

**Fig. 5 | Random transverse field Ising model on a chain. a** Relative error of the variational energy on a cluster of $N = 64$ sites, fixing $h_0 = 1.0$ on different train (left) and test (right) disorder realizations increasing the number of systems $\mathcal{R}$. The integer numbers (seeds) shown on the x-axis label different disorder realizations. Specifically, integers from 0 to 6 correspond to a set of seven different training disorder realizations, while those from 10 to 16 correspond to a set of test

realizations. **b** Spin-spin correlation function averaged over $\mathcal{R} = 1000$ disorder realizations at $h_0 = 1$. The dashed line represent the theoretical power law behaviour with exponent $\eta \approx 0.382$. **c** Square magnetization $m^2_{h_0}$ as a function of $h_0$. At fixed $h_0$ order parameter is obtained by averaging over $\mathcal{R} = 1000$ different disorder realizations. The numerically exact results are reported as comparison with solid lines.

constant at $M = 10000$. The analysis reveals that increasing $\mathcal{R}$ does not compromise the accuracy on the training seeds. In fact, even with an increase in training points to $\mathcal{R} = 1000$, we achieve highly accurate energy predictions while keeping the number of configurations per system relatively low, specifically $M/\mathcal{R} = 10$. More importantly, the generalization error on the test seeds (disorder realizations not encountered during training) systematically decreases when increasing $\mathcal{R}$. Notably, for $\mathcal{R} = 1000$, the relative errors of the training and test accuracies show the same order of magnitude, indicating that the FNQS has successfully learned how to combine the disorder couplings with the spin configurations to generate accurate amplitudes in the space of both physical configurations and couplings. We emphasize that the relative error for each disorder realization achieved by the FNQS is comparable to that obtained by training the same architecture on a single disorder realization (not reported here). This highlights the remarkable efficiency of the proposed method.

To assess the ability of FNQS to accurately predict disorder-averaged observables beyond energy, in Fig. 5b we show the average spin-spin correlation function at criticality:

$$C_{\text{av}}(r) = \frac{1}{N}\sum_{i=1}^{N}\int d\boldsymbol{\gamma}\,\mathcal{P}(\boldsymbol{\gamma})\left\langle \hat{S}^z_i \hat{S}^z_{i+r}\right\rangle_{\boldsymbol{\gamma}}. \qquad (8)$$

The average correlation function $C_{\text{av}}(r)$ is stochastically estimated by sampling $\mathcal{R} = 1000$ disorder realizations at $h_0 = 1$. Refer to Methods for

further details. We find good agreement with the theoretical critical scaling, characterized by the critical exponent $\eta = (3 - \sqrt{5})/2 \approx 0.382$, which is depicted as a dashed line in Fig. 5b. In Fig. 5c we measure the order parameter of the system as a function of $h_0$. In particular, for a fixed value of $h_0$, ranging from $h_0 = 0.4$ to $h_0 = 1.6$, we train a single FNQS over $\mathcal{R} = 1000$ distinct disorder realizations sampled for each $h_0$. After training, we estimate the square magnetization, defined as $m^2_{h_0} = 1/N\sum_{r=1}^{N}C_{\text{av}}(r)$. The variational results are in excellent agreement with numerically exact calculations across different system sizes, namely $N = 16, 32, 64$. Remarkably, achieving similar results with standard methods would require the optimization of 1000 independent simulations for each value of $h_0$, highlighting the efficiency and scalability of our approach. To provide a more stringent test of the accuracy of the predicted observables, in Fig. 6 we analyze the distribution of the square magnetization $m^2_{h_0}$ over a set of 1000 *test* disorder realizations *not* encountered during training. The comparison with exact results demonstrates excellent agreement for the different values of $h_0 = 0.4, 1.0$ and $1.6$, capturing not only the regions of high probability density but also the tails of the distributions with remarkable accuracy. In the inset of each panel of Fig. 6, we present correlation plots comparing the exact square magnetization with the FNQS predictions for disorder realizations *not* encountered during training. These plots further highlight the excellent agreement between the predictions and exact results, even for the most extreme and improbable values of the square magnetization.

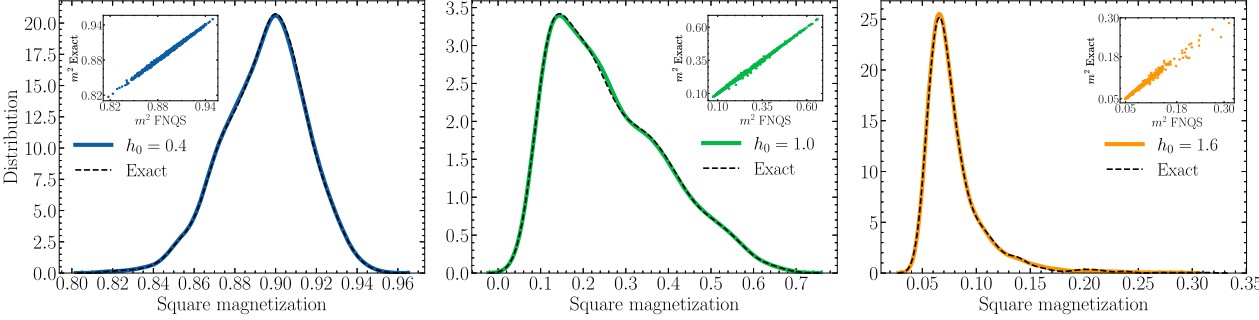

**Fig. 6 | Distributions of the square magnetization $m^2_{h_0}$ of the random transverse field Ising model.** The FNQS is trained on $\mathcal{R} = 1000$ independent disorder realizations and tested on a separate set of 1000 unseen realizations. The reported distributions correspond to the latter, with results presented on a chain of length $N = 32$ for three distinct disorder strengths: $h_0 = 0.4$, $h_0 = 1.0$, and $h_0 = 1.6$. For comparison, numerically exact results are included as black dashed lines. The insets of each panel illustrate the correlation between the exact squared magnetizations and the variational values predicted by the FNQS for unseen disorder realizations.

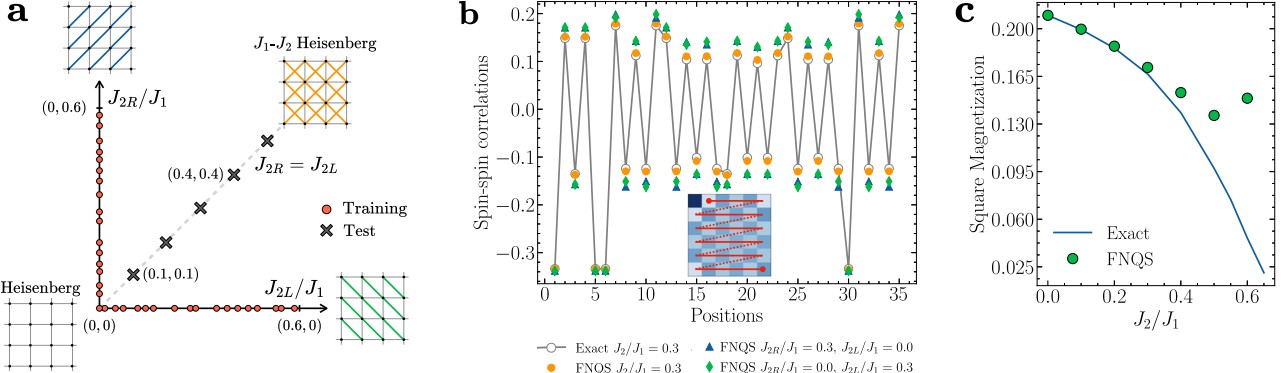

**Fig. 7 | Out of distribution generalization on the generalized $J_1$-$J_2$ Heisenberg model. a** Generalized $J_1$-$J_2$ Heisenberg model on a square lattice, with two couplings, $J_{2L}/J_1$ and $J_{2R}/J_1$. When $J_{2L} = J_{2R}$, the model reduces to the standard $J_1$-$J_2$ Heisenberg model. The FNQS is trained on configurations with frustration on only one diagonal and tested on configurations where both diagonals are frustrated. **b** Spin-spin correlation function on a 6 × 6 lattice of a FNQS trained on $\mathcal{R} = 1000$ different realizations of frustration affecting only one of the two diagonals of a square lattice (see Fig. 7 for a schematic representation). The model is tested on two in-distribution points $(J_{2L}/J_1, J_{2R}/J_1) = (0.0, 0.3)$ and $(0.3, 0.0)$, and on the out-of-distribution point $(J_{2L}/J_1, J_{2R}/J_1) = (0.3, 0.3)$, where both diagonals are frustrated. For reference, the exact results of the $J_1$-$J_2$ Heisenberg model at $J_2/J_1 = 0.3$ are also shown. Inset: The red line shows how the spin-spin correlations are ordered in the panel (**b**). **c** Out of distribution generalization of the square magnetization associated with Néel antiferromagnetic order for the FNQS (green circles) as a function of the frustration ratio $J_2/J_1$ on a 6 × 6 cluster. Exact results (blue line) are shown for reference.

## Out of distribution generalization

In this section, we investigate whether a FNQS trained on a restricted coupling domain can generalize to couplings outside this domain, namely for $\gamma \notin \text{Dom}\{\mathcal{P}(\gamma)\}$. In general, we do not expect this type of out of distribution generalization to succeed, as in any other machine learning approach. Nonetheless, we report here an example where this type of unconventional generalization is effective with some limitations.

Specifically, we consider the following Hamiltonian defined by generalizing the $J_1$-$J_2$ Heisenberg model on a $L \times L$ square lattice (with periodic boundary conditions)[10,60]:

$$\hat{H} = J_1 \sum_i \left( \hat{S}_i \cdot \hat{S}_{i+\hat{y}} + \hat{S}_i \cdot \hat{S}_{i+\hat{x}} \right) + \sum_i \left( J_{2R} \, \hat{S}_i \cdot \hat{S}_{i+\hat{x}+\hat{y}} + J_{2L} \, \hat{S}_i \cdot \hat{S}_{i+\hat{x}-\hat{y}} \right)$$

(9)

which depends on two distinct couplings, $J_{2L}/J_1$ and $J_{2R}/J_1$. When $J_{2L} = J_{2R} = 0$, the model reduces to the unfrustrated Heisenberg model on a square lattice[50]. Increasing $J_{2L}/J_1$ introduces frustration exclusively along the left diagonals of the square lattice, while increasing $J_{2R}/J_1$ does so along the right diagonals (see Fig. 7a). In the limiting cases where either $J_{2L} \neq 0$ and $J_{2R} = 0$, or vice versa, the model in Eq. (9)

corresponds to the Heisenberg model on the anisotropic triangular lattice[61–63].

To probe the generalization capability of the FNQS model, we design the following experiment (illustrated in Fig. 7a): the architecture is trained solely on Hamiltonians where frustration is present on only one diagonal at a time and then evaluated on Hamiltonians in which both diagonals are simultaneously frustrated. Specifically, the training data are sampled from a coupling distribution $\mathcal{P}(\gamma)$ defined exclusively on points of the form $(J_{2L}/J_1, 0)$ or $(0, J_{2R}/J_1)$, where only one of the two next-nearest-neighbor couplings is active, with $J_{2L}/J_1$ and $J_{2R}/J_1$ harvested from a uniform distribution defined on the interval $[0.0, 0.6]$. We then assess whether the resulting model can generalize to coupling configurations with $J_{2L} = J_{2R}$, which recover the $J_1$-$J_2$ Heisenberg model on a square lattice[10,60], where frustration is introduced symmetrically along both diagonals. Importantly, such test points lie outside the support of the training distribution $\mathcal{P}(\gamma)$, challenging the model's ability to extrapolate beyond its training regime. In Fig. 7b, we consider a 6 × 6 lattice and plot the spin-spin correlation function at two in-distribution points, $(J_{2L}/J_1, J_{2R}/J_1) = (0.0, 0.3)$ and $(0.3, 0.0)$, as well as at the out-of-distribution point $(0.3, 0.3)$. Remarkably, the FNQS accurately captures the enhanced frustration in the latter case, producing correlation functions that have lower amplitudes than in the

case with only one frustrated diagonal at a time, and in close agreement with exact results. This demonstrates the model's surprising ability to generalize beyond the support of the training distribution. However, it is important to emphasize that the accuracy of such generalization decreases as the distance from the training axis ($J_{2L} = 0$ or $J_{2R} = 0$) increases. Specifically, along the diagonal direction ($J_{2L} = J_{2R}$), the relative error on the ground-state energy remains very small in the Néel antiferromagnetic phase of the $J_1$-$J_2$ Heisenberg model, on the order of $10^{-5}$ for $J_2 = 0.1$, $10^{-4}$ for $J_2 = 0.3$, but increases substantially to order $10^{-2}$ for $J_2 = 0.5$ and $10^{-1}$ for $J_2 = 0.6$. This degradation in generalization performance is illustrated in Fig. 7c, which shows the behaviour of the Néel antiferromagnetic order parameter $m_{\text{Néel}}^2 = C(\pi, \pi)/N$, where $C(\mathbf{k}) = \sum_{\mathbf{r}} e^{i\mathbf{k}\cdot\mathbf{r}} \langle \hat{\mathbf{S}}_{\mathbf{0}} \cdot \hat{\mathbf{S}}_{\mathbf{r}} \rangle$ is the spin structure factor and $N$ is the total number of sites in the $J_1$-$J_2$ Heisenberg model ($J_{2L} = J_{2R}$).

## Discussion

We have demonstrated that a single neural-network architecture can be efficiently trained on multiple many-body quantum systems, yielding a variational state that generalizes to previously unseen coupling parameters. This approach enables the use of pre-trained states as starting points for specific investigations[25], similar to current practices in machine learning. To facilitate the adoption of this methodology, we have made FNQS models available through the Hugging Face Hub at https://huggingface.co/nqs-models, integrated with the `transformers` library[64] and providing simple interfaces for `NetKet`[65].

Several research directions emerge from this work. Specifically, we believe that in the near future, it will be possible to develop FNQS capable of treating all spin models with arbitrary two-body interactions in one and two dimensions. Achieving this ambitious goal will require a step-by-step approach and forms part of a broader long-term research program. Moreover, the extension to fermionic systems in second quantization[66,67] requires adapting the architecture while maintaining the core methodology. For molecular systems[68], the multimodal structure of FNQS could enable efficient computation of energy derivatives with respect to geometric parameters, providing access to atomic forces and equilibrium configurations. Beyond ground states, these foundation models could potentially facilitate the study of quantum dynamics by introducing explicit time-dependent variational states[69,70], particularly in large systems where traditional methods become intractable. These developments, combined with the public availability of pre-trained models, represent a step toward making advanced quantum many-body techniques more accessible to the broader physics community.

## Methods
### Expectation values
Given a set of operators $\hat{A}_{\boldsymbol{\gamma}}$ parametrized by the couplings $\boldsymbol{\gamma}$, its ensemble average over the distribution $\mathcal{P}(\boldsymbol{\gamma})$ is expressed as:

$$\mathcal{A}(\theta) = \int d\boldsymbol{\gamma} \mathcal{P}(\boldsymbol{\gamma}) \frac{\langle \psi_\theta(\boldsymbol{\gamma}) | \hat{A}_{\boldsymbol{\gamma}} | \psi_\theta(\boldsymbol{\gamma}) \rangle}{\langle \psi_\theta(\boldsymbol{\gamma}) | \psi_\theta(\boldsymbol{\gamma}) \rangle} . \tag{10}$$

This expectation value can be stochastically evaluated using a set of $\mathcal{R}$ couplings $\{\boldsymbol{\gamma}_1, \ldots, \boldsymbol{\gamma}_{\mathcal{R}}\}$ sampled from the probability distribution $\mathcal{P}(\boldsymbol{\gamma})$ as:

$$\mathcal{A}(\theta) \approx \frac{1}{\mathcal{R}} \sum_{k=1}^{\mathcal{R}} \frac{\langle \psi_\theta(\boldsymbol{\gamma}_k) | \hat{A}_{\boldsymbol{\gamma}_k} | \psi_\theta(\boldsymbol{\gamma}_k) \rangle}{\langle \psi_\theta(\boldsymbol{\gamma}_k) | \psi_\theta(\boldsymbol{\gamma}_k) \rangle} . \tag{11}$$

Each term in the sum of Eq. (11) can be rewritten as:

$$\frac{\langle \psi_\theta(\boldsymbol{\gamma}_k) | \hat{A}_{\boldsymbol{\gamma}_k} | \psi_\theta(\boldsymbol{\gamma}_k) \rangle}{\langle \psi_\theta(\boldsymbol{\gamma}_k) | \psi_\theta(\boldsymbol{\gamma}_k) \rangle} = \sum_{\boldsymbol{\sigma}} p_\theta(\boldsymbol{\sigma} | \boldsymbol{\gamma}_k) \frac{\langle \boldsymbol{\sigma} | \hat{A}_{\boldsymbol{\gamma}_k} | \psi_\theta(\boldsymbol{\gamma}_k) \rangle}{\langle \boldsymbol{\sigma} | \psi_\theta(\boldsymbol{\gamma}_k) \rangle} . \tag{12}$$

where we have defined the probability distribution $p_\theta(\boldsymbol{\sigma}|\boldsymbol{\gamma}_k) = |\psi_\theta(\boldsymbol{\sigma}|\boldsymbol{\gamma}_k)|^2/\langle\psi_\theta(\boldsymbol{\gamma}_k)|\psi_\theta(\boldsymbol{\gamma}_k)\rangle$. In the Variational Monte Carlo (VMC) framework[17], this expectation value can be further estimated stochastically over a set of $M_k$ physical configurations $\{\boldsymbol{\sigma}_1, \ldots, \boldsymbol{\sigma}_{M_k}\}$ sampled according to the probability distribution $p_\theta(\boldsymbol{\sigma}|\boldsymbol{\gamma}_k)$:

$$\bar{A}_k = \frac{1}{M_k} \sum_{j=1}^{M_k} \frac{\langle \boldsymbol{\sigma}_j | \hat{A}_{\boldsymbol{\gamma}_k} | \psi_\theta(\boldsymbol{\gamma}_k) \rangle}{\langle \boldsymbol{\sigma}_j | \psi_\theta(\boldsymbol{\gamma}_k) \rangle} . \tag{13}$$

In the calculations performed in this work, we set an equal number of samples for each system, $M_k = M/\mathcal{R}$, independent of $k$, where $M$ is the total number of samples in the extended space of all systems. See to ref. 17 for further details on the VMC framework.

### Stochastic reconfiguration for multiple systems
A contribution of this work is the generalization of the Stochastic Reconfiguration (SR) method[15–17] to optimize a variational wave function that approximates ground states of an ensemble of Hamiltonians, thus minimizing the loss in Eq. (1). Unlike the standard single-system setting, the SR equation here is obtained by minimizing the ensemble-averaged fidelity between the exact imaginary-time evolution and its variational approximation, employing the Time-Dependent Variational Principle (TDVP)[71].

In the single-system case, characterized by the coupling parameters $\boldsymbol{\gamma}$, the fidelity between the state evolved in imaginary time under the exact Hamiltonian for a time-step $\varepsilon$, namely $e^{-\varepsilon \hat{H}_{\boldsymbol{\gamma}}} | \psi_{\theta(\tau)}(\boldsymbol{\gamma}) \rangle$, and the corresponding variationally evolved state $| \psi_{\theta(\tau) + \varepsilon \dot{\theta}(\tau)}(\boldsymbol{\gamma}) \rangle$ is defined as:

$$f^2(\boldsymbol{\gamma}) = \frac{|\langle \psi_\theta(\boldsymbol{\gamma}) | e^{-\varepsilon \hat{H}_{\boldsymbol{\gamma}}} | \psi_{\theta + \varepsilon \dot{\theta}}(\boldsymbol{\gamma}) \rangle|^2}{\langle \psi_\theta(\boldsymbol{\gamma}) | e^{-2\varepsilon \hat{H}_{\boldsymbol{\gamma}}} | \psi_\theta(\boldsymbol{\gamma}) \rangle \langle \psi_{\theta + \varepsilon \dot{\theta}}(\boldsymbol{\gamma}) | \psi_{\theta + \varepsilon \dot{\theta}}(\boldsymbol{\gamma}) \rangle} . \tag{14}$$

Here, $\dot{\theta}_\alpha(\tau)$ denotes the derivative of the $\alpha$-th variational parameter with respect to imaginary time $\tau$, with $\alpha = 1, \ldots, P$ and $P$ the total number of parameters. For simplicity in the notation, in the following, we omit the explicit time dependence of the variational parameters. To generalize to an ensemble of systems, we define the global fidelity as the ensemble average of the fidelity over the distribution $\mathcal{P}(\boldsymbol{\gamma})$ as $\mathcal{F}^2 = \int d\boldsymbol{\gamma} \mathcal{P}(\boldsymbol{\gamma}) f^2(\boldsymbol{\gamma})$. Assuming real-valued variational parameters and expanding to second order in $\varepsilon$, we obtain:

$$\mathcal{F}^2 \approx 1 - \varepsilon^2 \left[ \dot{\boldsymbol{\theta}}^T \mathcal{G} + \dot{\boldsymbol{\theta}}^T \mathcal{S} \dot{\boldsymbol{\theta}} + \int d\boldsymbol{\gamma} \mathcal{P}(\boldsymbol{\gamma}) \text{Var}(\hat{H}_{\boldsymbol{\gamma}}) \right] , \tag{15}$$

where $\mathcal{G}_\alpha = \partial \mathcal{L}(\theta)/\partial \theta_\alpha$ is the gradient of the loss and is defined as the ensemble average $\mathcal{G}_\alpha = \int d\boldsymbol{\gamma} \mathcal{P}(\boldsymbol{\gamma}) G_\alpha(\boldsymbol{\gamma})$, with $G_\alpha(\boldsymbol{\gamma}) = \partial \langle \hat{H}_{\boldsymbol{\gamma}} \rangle_{\boldsymbol{\gamma}}/\partial \theta_\alpha$ which can be rewritten as:

$$G_\alpha(\boldsymbol{\gamma}) = 2\Re \left\{ \langle \hat{H}_{\boldsymbol{\gamma}} \hat{O}_{\boldsymbol{\gamma}, \alpha} \rangle_{\boldsymbol{\gamma}} - \langle \hat{H}_{\boldsymbol{\gamma}} \rangle_{\boldsymbol{\gamma}} \langle \hat{O}_{\boldsymbol{\gamma}, \alpha} \rangle_{\boldsymbol{\gamma}} \right\} . \tag{16}$$

In the last equation, $\hat{O}_{\boldsymbol{\gamma}, \alpha}$ is a diagonal operator in the computational basis of the system characterized by couplings $\boldsymbol{\gamma}$, whose matrix elements are defined as $\langle \boldsymbol{\sigma} | \hat{O}_{\boldsymbol{\gamma}, \alpha} | \boldsymbol{\sigma}' \rangle = \delta_{\boldsymbol{\sigma}, \boldsymbol{\sigma}'} \partial \text{Log}[\psi_\theta(\boldsymbol{\sigma}|\boldsymbol{\gamma})]/\partial \theta_\alpha$. Analogously, the matrix $\mathcal{S}$ is defined as $\mathcal{S}_{\alpha, \beta} = \int d\boldsymbol{\gamma} \mathcal{P}(\boldsymbol{\gamma}) S_{\alpha, \beta}(\boldsymbol{\gamma})$, with $S(\boldsymbol{\gamma})$ being the real part of the quantum geometric tensor defined as:

$$S_{\alpha\beta}(\boldsymbol{\gamma}) = \Re \left\{ \langle \hat{O}_{\boldsymbol{\gamma}, \alpha}^\dagger \hat{O}_{\boldsymbol{\gamma}, \beta} \rangle_{\boldsymbol{\gamma}} - \langle \hat{O}_{\boldsymbol{\gamma}, \alpha}^\dagger \rangle_{\boldsymbol{\gamma}} \langle \hat{O}_{\boldsymbol{\gamma}, \beta} \rangle_{\boldsymbol{\gamma}} \right\} . \tag{17}$$

Importantly, the matrix $\mathcal{S}$ is constructed as an ensemble average of the matrices $S_{\alpha\beta}(\boldsymbol{\gamma})$ of the individual systems, weighted by the probability distribution $\mathcal{P}(\boldsymbol{\gamma})$. In the absence of this weighting, $\mathcal{S}$ would reduce to an unweighted integral, leading to large statistical fluctuations as the number of systems increases and potentially diverging variances in its elements.

The TDVP equations for the ensemble are obtained by maximizing the global fidelity in Eq. (15) with respect to $\dot{\boldsymbol{\theta}}$, leading to the linear system $\mathcal{S}\dot{\boldsymbol{\theta}} = -\frac{1}{2}\mathcal{G}$. This differential equation can then be integrated numerically. In ground-state applications, it is common to employ the simple Euler scheme, which approximates the time derivative as $\dot{\theta}_\alpha(\tau) \approx [\theta(\tau + \eta) - \theta(\tau)]/\eta$. Here, $\tau$ denotes the imaginary time parameterizing the dynamics, and $\eta$ is the integration time step used to discretize the evolution. Based on these results, the SR updates are conventionally defined by removing the factor of $1/2$[72,73] leading to $\boldsymbol{\delta\theta} = -\eta\mathcal{S}^{-1}\mathcal{G}$, where we have defined $\delta\theta_\alpha = \theta_\alpha(\tau + \eta) - \theta_\alpha(\tau)$. It is important to consider that the matrix $\mathcal{S}$ may possess extremely small or even negligible eigenvalues. As a result, directly computing its inverse can lead to numerical instabilities[17]. To mitigate these potential issues, we adopt a regularized update scheme of the form:

$$\boldsymbol{\delta\theta} = -\eta(\mathcal{S} + \lambda\mathbb{I}_P)^{-1}\mathcal{G}, \qquad (18)$$

where $\eta$ acts as a learning rate controlling the update magnitude during the optimization process, and $\lambda > 0$ is a regularization parameter introduced to ensure the invertibility and numerical stability of the matrix $\mathcal{S}$ matrix[17,19]. The same linear algebra formula introduced in ref. 19 can be employed to enhance efficiency when the number of variational parameters $P$ significantly exceeds the number of samples $M$ used for stochastic estimations, as is typical for FNQS.

### Generalized fidelity susceptibility

A rigorous approach for the unsupervised detection of quantum phase transitions involves measuring the fidelity susceptibility[31]. Consider a system described by the Hamiltonian $\hat{H}_{\boldsymbol{\gamma}}$ characterized by $N_c$ couplings $\boldsymbol{\gamma} = (\gamma^{(1)}, \gamma^{(2)}, \ldots, \gamma^{(N_c)})$. First, we introduce the fidelity defined as:

$$F^2(\boldsymbol{\gamma}, \boldsymbol{\varepsilon}) = \frac{|\langle\psi_\theta(\boldsymbol{\gamma})|\psi_\theta(\boldsymbol{\gamma} + \boldsymbol{\varepsilon})\rangle|^2}{\langle\psi_\theta(\boldsymbol{\gamma})|\psi_\theta(\boldsymbol{\gamma})\rangle\langle\psi_\theta(\boldsymbol{\gamma} + \boldsymbol{\varepsilon})|\psi_\theta(\boldsymbol{\gamma} + \boldsymbol{\varepsilon})\rangle}. \qquad (19)$$

It quantifies the overlap between two quantum states on the manifold of the couplings $\boldsymbol{\gamma}$ and it shows a dip in correspondence with a quantum phase transition[31–33]. Expanding the fidelity in a Taylor series around $\boldsymbol{\varepsilon} = 0$, we have:

$$F^2(\boldsymbol{\gamma}, \boldsymbol{\varepsilon}) = 1 - \sum_{i,j=1}^{N_c} \varepsilon_i\varepsilon_j\chi_{ij}(\boldsymbol{\gamma}) + O(|\boldsymbol{\varepsilon}|^3), \qquad (20)$$

where the *generalized fidelity susceptibility* $\chi_{ij}(\boldsymbol{\gamma})$, a $N_c \times N_c$ symmetric positive-definite matrix, represents the leading non-zero contribution. It is easy to show that it can be obtained as:

$$\chi_{ij}(\boldsymbol{\gamma}) = -\left.\frac{\partial^2 \ln F(\boldsymbol{\gamma}, \boldsymbol{\varepsilon})}{\partial\varepsilon_i\partial\varepsilon_j}\right|_{\boldsymbol{\varepsilon}=0}. \qquad (21)$$

In the case of a single coupling ($N_c = 1$), the tensor $\chi_{ij}(\boldsymbol{\gamma})$ simplifies to a scalar function, which peaks at the phase transition and diverges in the thermodynamic limit. However, even in this simpler case, computing the fidelity susceptibility is difficult. Standard approaches require evaluating the ground state for each coupling value, computing the fidelity, and then using finite-difference methods to estimate its second derivative [see Eq. (21)]. However, the fidelity becomes exponentially small as the system size increases, making the procedure numerically challenging. As a result, fidelity susceptibility is typically computed via exact diagonalization on small clusters or tensor network methods in one-dimensional systems[37]. Efficient algorithms based on Quantum Monte Carlo methods have been proposed to address this challenge, but they are limited to systems with positive-definite ground states[31].

In this work, we propose an alternative approach that overcomes these limitations. The matrix $\chi_{ij}(\boldsymbol{\gamma})$ in Eq. (21) can be equivalently computed as the real part of the *quantum geometric tensor* with respect to couplings $\boldsymbol{\gamma}$[32,33] as:

$$\chi_{ij}(\boldsymbol{\gamma}) = \Re\left\{\langle\hat{\mathcal{O}}_{\boldsymbol{\gamma},i}^\dagger\hat{\mathcal{O}}_{\boldsymbol{\gamma},j}\rangle_{\boldsymbol{\gamma}} - \langle\hat{\mathcal{O}}_{\boldsymbol{\gamma},i}^\dagger\rangle_{\boldsymbol{\gamma}}\langle\hat{\mathcal{O}}_{\boldsymbol{\gamma},j}\rangle_{\boldsymbol{\gamma}}\right\}. \qquad (22)$$

The operators $\hat{\mathcal{O}}_{\boldsymbol{\gamma},i}$ are diagonal in the computational basis whose matrix elements are defined as $\langle\boldsymbol{\sigma}|\hat{\mathcal{O}}_{\boldsymbol{\gamma},i}|\boldsymbol{\sigma}'\rangle = \delta_{\sigma,\sigma'}\partial \text{Log}[\psi_\theta(\boldsymbol{\sigma}|\boldsymbol{\gamma})]/\partial\gamma^{(i)}$, where $\gamma^{(i)}$ is the $i$-th component of the coupling vector $\boldsymbol{\gamma}$. By exploiting the multimodal nature of the FNQS wave function, it is possible to compute the derivatives of the amplitudes with respect to the Hamiltonian couplings, a highly non-trivial quantity that is inaccessible for standard variational states optimized on a single value of the couplings. As a result, for FNQS, the quantum geometric tensor in Eq. (22) can be directly computed using automatic differentiation techniques, bypassing the need to explicitly calculate the fidelity.

We emphasize that identifying quantum phase transitions without prior knowledge of order parameters is a challenging task, and existing state-of-the-art methods have notable limitations that hinder their applicability in complicated scenarios. For instance, supervised approaches[74] require prior knowledge of the different phases, while unsupervised techniques are generally restricted to models with a single physical coupling[75] or rely on quantum tomography, which is typically computationally demanding[76,77]. All these limitations are overcome by our approach, which extends the computation of fidelity susceptibility[31] to general physical models with multiple couplings.

## Data availability
The data that support the findings of this study are available from the corresponding author upon request.

## Code availability
The architectures trained in this paper are publicly available at https://huggingface.co/nqs-models, along with examples for implementing these neural networks in NetKet[65].

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

## Acknowledgements

We thank S. Amodio for preparing Fig. 1. R.R and L.L.V. acknowledge the CINECA award under the ISCRA initiative for the availability of high-performance computing resources and support. The work of AS was funded by the European Union–NextGenerationEU under the project NRRP "National Centre for HPC, Big Data and Quantum Computing (HPC)" CN00000013 (CUP D43C22001240001) [MUR Decree n. 341–15/03/2022] – Cascade Call launched by SPOKE 10 POLIMI: "CQEB" project, and from the National Recovery and Resilience Plan (NRRP), Mission 4 Component 2 Investment 1.3 funded by the European Union NextGenerationEU, National Quantum Science and Technology Institute (NQSTI), PE00000023, Concession Decree No. 1564 of 11.10.2022 adopted by the Italian Ministry of Research, CUP J97G22000390007. This work was supported by the Swiss National Science Foundation under Grant No. 200021_200336.

## Author contributions

R.R. and L.L.V. performed the numerical simulations. R.R., L.L.V., F.B., A.S., A.L., G.C. devised the framework and wrote the manuscript.

## Competing interests
The authors declare no competing interests.
