## [Transparent Peer Review file · Nature Communications]

Foundation Neural-Networks Quantum States as a Unified Ansatz for Multiple Hamiltonians

Corresponding Author: Dr Riccardo Rende

Version 0:

Reviewer comments:

Reviewer #1

(Remarks to the Author)

This manuscript introduces the Foundation Neural-Network Quantum States (FNQS) framework—a novel approach that adapts the ideas behind modern foundation models to the simulation of quantum many-body systems. The authors propose a single, versatile neural-network architecture that accepts multimodal inputs (both physical configurations and Hamiltonian parameters) and can be trained simultaneously over many quantum systems.

To my knowledge, the results presented in this work are novel and noteworthy. One highlight of the paper is the computation of the generalized fidelity susceptibility, which enables unsupervised detection of quantum phase transitions without prior knowledge of order parameters. I recommend the paper published with revisions.

Here are some detailed questions and comments:

1. The title “Foundation Neural-Network Quantum States” may overstate the generality of the approach. Foundation models are typically trained on vast, diverse datasets to address a wide range of use cases. In contrast, the reported results are based on specific systems and tasks. Could the authors elaborate on how FNQS would handle more diverse datasets or extend to a broader class of quantum systems?
2. Sec. III A shows that FNQS can yield good predictions even with very few training data points, with Fig. 2b’s inset demonstrating a reasonable interpolation between two training points. Given the high-dimensional nature of the Hilbert space and the complexity introduced by the phase transition at $h = 1$, it is surprising that such limited data can lead to physically meaningful interpolation. What features of the FNQS architecture or training procedure facilitate this behavior? Is there an inherent bias or physical prior encoded in the network that aids in capturing the underlying physics?
3. The manuscript would benefit from additional details on how the hyperparameters were chosen, particularly regarding the architecture and size of the neural network. For instance, is there a scaling relationship between the number of parameters, the number of qubits, and the model’s performance? Given the known sensitivity of large language models to network size, a discussion on whether similar scaling laws apply here would be valuable.

A few minor comments:

1. The first half of the paper appears to follow methodologies established in [26]. It would be helpful if the authors could clearly delineate their original contributions from those based on established results.
2. The insets in Fig. 2 are currently too small to be legible.

Reviewer #2

(Remarks to the Author)

This manuscript introduces “Foundation Neural-Network Quantum States”, an extension to the variational Monte Carlo framework with neural network quantum states. The key idea is to train a single, versatile neural network to approximate the ground states of a family of parameterized Hamiltonians. The network takes both the physical configuration basis and the Hamiltonian parameters as inputs and outputs the amplitude. The author demonstrate the numerical approach on the 1D TFIM, 2D J1-J2-J3 Heisenberg model, and the random field TFIM. Applications beyond conventional nqs-based VMC are shown including efficient calculation of disorder-averaged quantities and the unsupervised detection of phase transitions, which can be computed efficiently using automatic differentiation.

The paper presents a timely and potentially impactful direction for applying advanced machine learning concepts to quantum many-body problems. The core idea of training a single NQS model conditioned on Hamiltonian parameters to

learn ground state properties across a parameter range, though not that novel, is compelling and offers potential practical advantages, particularly for disordered systems. The method for unsupervised probing of phase transitions is also very smart designed and inspiring. I also appreciate the open source practice and the new open source paradigm on huggingface.

There are several aspects that require further clarification and investigation. After addressing the following comments, I believe this work is acceptable on Nature Communications.

1. Stochastic Reconfiguration Generalization: While technically necessary for the proposed framework, the generalization of SR (Eq. 4-6) is relatively straightforward and barely an "original contribution". It should be presented as an enabling technical step rather than a primary contribution of the work itself.

2. Generalization Capability: Generalization is central to foundation models. The current manuscript demonstrates generalization primarily in two forms: (i) interpolation within the training parameter range and (ii) generalization to unseen disorder instances drawn from the same distribution. A more stringent test, characteristic of foundation models, would be extrapolation – testing the model significantly outside the parameter range it was trained on. For instance, if trained on TFIM with h in $[0.8, 1.2]$, how does it perform at $h=0.5$ or $h=1.5$? Or even qualitatively different phases if applicable. The authors should perform such extrapolation tests and discuss the limitations and implications of the model's generalization capabilities more carefully.

3. Inputting Hamiltonian Parameters: The current strategies handle $O(1)$ and $O(N)$ couplings. However, Hamiltonians can have more complex structures (e.g., non-local interactions, varying connectivity) not easily captured by simple vector concatenation or patch-wise splitting. A discussion on limitations and potential alternative representations would strengthen the paper. For example, for a general parameterized Hamiltonian represented as a graph, is there a suitable multimodal architecture to handle such input? (similar scenario occurs for LLM where images can be fed into the model together with texts)

4. Related Work: The authors mention previous attempts [26-29]. It would be beneficial to slightly expand on why FNQS overcomes the limitations mentioned. Furthermore, while distinct from NQS, there is previous work on variational quantum states that shares the same foundation model philosophy of this manuscript, where optimized parameterized quantum circuits/ground state wavefunctions conditioned on problem Hamiltonian parameters are learnt by a neural network (<https://journals.aps.org/prapplied/abstract/10.1103/PhysRevApplied.21.014053>). Namely, in the current manuscript, a ground state representation in the form of NQS conditioned on Hamiltonian parameter is learnt, while in the previous work, a ground state representation in the form of PQC conditioned on Hamiltonian parameter is learnt. The authors should cite this to place the current manuscript in a broader context.

5. Cost Analysis: Consider adding a brief discussion on the computational cost scaling: How does training an FNQS for R systems with total batch size M compare to training R independent NQS models each with batch size M/R ?

Reviewer #3

(Remarks to the Author)

The authors of this manuscript modify the well-established neural-network quantum states approach by using foundation models and network architectures. With this modification, the authors are able to consider spin configurations and Hamiltonian physical couplings when finding quantum state representations with the foundation model. This way, the approach can generalize to represent quantum states for previously unseen Hamiltonian parameters. They further show that with this approach, the fidelity susceptibility can be evaluated, which can be used to discover phase transitions without including prior knowledge.

With their work, the authors address a rapidly evolving field and provide significant advances in studying quantum many-body systems with the potential for applications in different systems. The proposed generalizability of this method suggests impactful advances in modelling phase diagrams of not well understood systems, which is of significant relevance to various disciplines of physics and beyond. While the claims made in the paper are groundbreaking and the results are legit, the authors need to address the following points in a revised manuscript before it can be considered for publication in Nature Communications:

1. In the abstract, the authors claim that "FNQS can generalize to physical Hamiltonians beyond those encountered during training, offering a unified framework adaptable to various quantum systems and tasks". However, in the results section, the authors only prove that the method generalizes to parameter choices not encountered during training, but using the same Hamiltonian as during the training. I do not see justification for this claim in the work and the authors should either reformulate this statement or show evidence that the introduced method does generalize to different Hamiltonians. In the manuscript, the authors use similar expressions multiple times (family of Hamiltonians, ensemble of Hamiltonians,...), all these expressions should be addressed.

2. I don't understand what the most right panel in Figure 1 is showing us. I also don't see it explained in the figure caption. The authors should comment on this.

3. The methods section requires some more information. Specifically:

- "Consider a family of Hamiltonians, denoted by H_γ ": What defines a family of Hamiltonians? How are they related? Looking at the results section, they seem to be the same Hamiltonian with different parameter choices. In this case I would find the expression "family" confusing.

- The authors say that their goal is to find an approximation of the ground state of an ensemble of Hamiltonians using a variational wave function which explicitly depends on the physical couplings and on a unique set of variational parameters for all the Hamiltonians. Why is this set of variational parameters unique? How is this guaranteed?

- How is the probability distribution $P(\gamma)$ over the Hamiltonian couplings defined? How can it be determined in a specific case?
- What are the coupling states? How are they defined?
- The coupling states satisfy $\langle \gamma | \gamma' \rangle = \delta(\gamma - \gamma')$, but it is said that γ can be discrete or continuous. In the second last paragraph before II.A, $P(\gamma)$ is even defined with a sum over γ_k , showing that they are discrete. For the discrete case, the usage of the Dirac delta function $\delta(\gamma - \gamma')$ does not make sense and should be replaced by the Kronecker $\delta_{\{\gamma, \gamma'\}}$.
- In the second last line of the left column on page 3, the authors define the diagonal operator in the computational basis of the system characterized by couplings γ . But the definition is given for $O_{\{\alpha\}}(\sigma, \gamma)$ and in eq. (5) the operator $\hat{O}_{\{\alpha, \gamma\}}$ appears. Are those the same operators? Where did the dependence on σ go? Or how does one relate to the other? The same expressions appear around eq. (9).

4. The axis ticks and labels, as well as the legends of the insets are way too small in Figs 2 and 5. They are unreadable in a printed version and should be increased.

5. In the presented results, the models are always trained closely around phase transitions, including training points in all different phases. While it is great to see that the model then generalizes well, it does not provide evidence for application in broader systems. How does the generalizability behave if the model is only trained on one side of the phase transition? What about training only on points far away from the transition, deep in the phases? Thinking about general physical systems, where the phase diagram is not known, I would assume that the training points can lie somewhere in the phase diagram and they are not necessarily close to a transition and do not necessarily cover all existing phases. A study on more general choices of training points should be added to this work to make it more consistent.

6. On page 5, end of right column, the authors say that the generated fidelity susceptibility data "can be used to study the collapse of the curves, enabling the extraction of the critical exponents of the Ising model." Have the authors studied this collapse and shown that the expected exponents can be extracted? If such a statement is made, I would expect to see the results to support the validity of the produced data.

7. In Fig. 3, panel a, the color bar is clipped to [0, 0.5]. Why is this clipping applied? What would be the plot without clipping?

8. How does this method scale to larger system sizes? The authors study up to 10x10 qubits, but do not comment on the computational costs. How expensive are the shown simulations and what kind of scaling is expected when going to larger systems? Especially when studying phase diagrams, which the authors propose as a possible use case of their method due to the generalizability, it is important to study the scaling behaviour to large system sizes. It would thus be essential to know if this method can be efficiently applied to large-scale systems.

9. In Fig. 4, right plot of panel a, the x-axis starts counting the seeds at 10. Why is this the case?

10. After eq. (14), the authors set $J=1/e$. What is e and why is this specific choice made?

Version 1:

Reviewer comments:

Reviewer #1

(Remarks to the Author)

I thank the authors for accepting my suggestions and for the substantial improvements made to the manuscript. Most of my concerns have been addressed, and I believe the paper is now suitable for publication after the resolution of a few additional issues:

1. In their response to Referee 2's Comment 2, the authors demonstrate the limitations of out-of-distribution generalization using the TFIM model. However, this result does not appear in the revised manuscript. I recommend including this experiment in Appendix F alongside the J1-J2 model. The observed limitations are reasonable and, in my view, as informative as the successes. I also suggest adding a brief sentence in the conclusions to remind readers that FNQS cannot be expected to extrapolate across phase boundaries without additional training.
2. There is a small typo in the caption of Fig. 1: "forth panel" should be corrected to "fourth panel."
3. The figures are not inserted properly in the provided pdf.

Reviewer #2

(Remarks to the Author)

The authors have fully addressed my comments, and I think the manuscript can be published on NC in the current form.

Reviewer #3

(Remarks to the Author)

The authors have addressed all points raised by the referees in good detail and have significantly improved the quality and clarification of the manuscript.

As was raised by the majority of the referees, the initial manuscript was quite misleading about the capabilities of the introduced method. The authors wrote the manuscript in a way that made it sound like the algorithm would generalize well over several regimes of single Hamiltonians and at some points even over various Hamiltonians. The authors clarified in their reply that this is not the case and adapted the manuscript accordingly. However, given the fact that this approach does not generalize more than existing methods (such as transformer-based neural network quantum states <https://arxiv.org/pdf/2405.21052>, <https://journals.aps.org/prb/abstract/10.1103/PhysRevB.107.075147>), I am unsure if the manuscript meets the requirements for publication in Nature Communications, as it significantly reduces the strength of the presented results.

Response to the Referee 1

Discussion — This manuscript introduces the Foundation Neural-Network Quantum States (FNQS) framework—a novel approach that adapts the ideas behind modern foundation models to the simulation of quantum many-body systems. The authors propose a single, versatile neural-network architecture that accepts multimodal inputs (both physical configurations and Hamiltonian parameters) and can be trained simultaneously over many quantum systems. To my knowledge, the results presented in this work are novel and noteworthy. One highlight of the paper is the computation of the generalized fidelity susceptibility, which enables unsupervised detection of quantum phase transitions without prior knowledge of order parameters. I recommend the paper published with revisions.

We thank Referee 1 for their reply. We hereby address in detail the points raised in their report.

Comment 1 — The title “Foundation Neural-Network Quantum States” may overstate the generality of the approach. Foundation models are typically trained on vast, diverse datasets to address a wide range of use cases. In contrast, the reported results are based on specific systems and tasks. Could the authors elaborate on how FNQS would handle more diverse datasets or extend to a broader class of quantum systems?

Reply: In this work, we set up the key conceptual and technical aspects that, we believe, are important ingredients to develop foundation neural-network quantum states. We notice that the applications presented here are already unprecedented in terms of the variety and complexity of models captured by a single neural network wave function. However, we agree that to further extend the framework to wider classes of Hamiltonians, it will be crucial to significantly extend the training set. This is part of our longer-term research programme, which will require substantial computational resources. In view of these considerations, we decided to modify the title as “Foundation Neural-Networks Quantum States as a Unified Ansatz for Multiple Hamiltonians”. Specifically, we believe that in the near future it will be possible to have FNQS that are able to treat all spin models (with arbitrary two-body interactions) in one and two dimensions. We have also added a comment on this aspect in the Conclusions.

Comment 2 — Sec. III A shows that FNQS can yield good predictions even with very few training data points, with Fig. 2b’s inset demonstrating a reasonable interpolation between two training points. Given the high-dimensional nature of the Hilbert space and the complexity introduced by the phase transition at $h = 1$, it is surprising that such limited data can lead to physically meaningful interpolation. What features of the FNQS architecture or training procedure facilitate this behavior? Is there an inherent bias or physical prior encoded in the network that aids in capturing the underlying physics?

Reply: We thank the reviewer for the thoughtful question. The key ingredient of our procedure is considering the Hamiltonian couplings as input parameters, at the same level of the physical configurations. The ability of the FNQS to interpolate meaningfully between different phases with limited data stems primarily by using a standard Vision Transformer

on this extended input data space. In particular, the multi-head attention mechanism plays a crucial role: each attention head can, in principle, specialize in capturing different features associated with distinct phases of the system, and these features “blend” the representations of the spin configurations with the representations of the couplings. This allows the network to efficiently learn a compact description of the various phases present. Furthermore, the all-to-all connectivity inherent in the attention mechanism enables the network to flexibly model long-range correlations across the system, which are essential for describing critical behaviour. Importantly, we believe the observed interpolation arises from the fact that the two training points were selected on opposite sides of the phase transition, each representative of distinct phases. The fact that the wave function is exposed to the physical properties of both regimes appears to encourage the model to interpolate smoothly between the two phases, rather than overfitting to isolated features of each. We added a comment in the main text.

Comment 3 — The manuscript would benefit from additional details on how the hyperparameters were chosen, particularly regarding the architecture and size of the neural network. For instance, is there a scaling relationship between the number of parameters, the number of qubits, and the model’s performance? Given the known sensitivity of large language models to network size, a discussion on whether similar scaling laws apply here would be valuable.

Reply: We added details about the performance of the model when increasing the number of parameters and for different system sizes in the Section “Systematic energy improvement” of the Appendix.

Minor Comment 1 — The first half of the paper appears to follow methodologies established in [26]. It would be helpful if the authors could clearly delineate their original contributions from those based on established results.

Reply: We have now better clarified the original contributions of our work in the Introduction.

Minor Comment 2 — The insets in Fig. 2 are currently too small to be legible.

Reply: We have modified the plots.

Response to Referee 2

Discussion — This manuscript introduces “Foundation Neural-Network Quantum States”, an extension to the variational Monte Carlo framework with neural network quantum states. The key idea is to train a single, versatile neural network to approximate the ground states of a family of parameterized Hamiltonians. The network takes both the physical configuration basis and the Hamiltonian parameters as inputs and outputs the amplitude. The author demonstrate the numerical approach on the 1D TFIM, 2D J_1 - J_2 - J_3 Heisenberg model, and

the random field TFIM. Applications beyond conventional nqs-based VMC are shown including efficient calculation of disorder-averaged quantities and the unsupervised detection of phase transitions, which can be computed efficiently using automatic differentiation.

The paper presents a timely and potentially impactful direction for applying advanced machine learning concepts to quantum many-body problems. The core idea of training a single NQS model conditioned on Hamiltonian parameters to learn ground state properties across a parameter range, though not that novel, is compelling and offers potential practical advantages, particularly for disordered systems. The method for unsupervised probing of phase transitions is also very smart designed and inspiring. I also appreciate the open source practice and the new open source paradigm on huggingface.

There are several aspects that require further clarification and investigation. After addressing the following comments, I believe this work is acceptable on Nature Communications.

We thank Referee 2 for their reply. We hereby address in detail the points raised in their report.

Comment 1 — Stochastic Reconfiguration Generalization: While technically necessary for the proposed framework, the generalization of SR (Eq. 4-6) is relatively straightforward and barely an “original contribution”. It should be presented as an enabling technical step rather than a primary contribution of the work itself.

Reply: In response to the comments by Referee 3 regarding the coupling states $|\gamma\rangle$, which appeared in the first version of our manuscript, we have substantially revised the text to avoid introducing these states. Instead, we now reformulate the framework using the concept of an ensemble average. Accordingly, we have revised the section on the generalization of the Stochastic Reconfiguration (SR) method, deriving it as the minimizer of the ensemble-averaged infidelity between the variational state and the exactly evolved state in imaginary time. This approach clarifies the key distinction from the standard single-system case, which we now highlight both in the main text and in the Appendix (“Derivation of the Stochastic Reconfiguration equation for multiple systems”). To avoid possible confusion, we have also replaced the term “original contribution” with the more neutral “contribution”.

Comment 2 — Generalization Capability: Generalization is central to foundation models. The current manuscript demonstrates generalization primarily in two forms: (i) interpolation within the training parameter range and (ii) generalization to unseen disorder instances drawn from the same distribution. A more stringent test, characteristic of foundation models, would be extrapolation – testing the model significantly outside the parameter range it was trained on. For instance, if trained on TFIM with h in $[0.8, 1.2]$, how does it perform at $h = 0.5$ or $h = 1.5$? Or even qualitatively different phases if applicable. The authors should perform such extrapolation tests and discuss the limitations and implications of the model’s generalization capabilities more carefully.

Reply: We appreciate the reviewer’s insightful remark. We would like to clarify that our approach, like any other Machine Learning approach, is intrinsically limited to instances that

are represented by the probability distribution underlying a given training set of problems. It would be inconceivable to use a foundation language model trained, say, on English, Spanish, and Chinese for a downstream task in which the prompts are provided in, say, Hungarian. Not surprisingly, considering the TFIM and by training the FNQS *only* in the disordered phase ($h > 1.2$), the model fails to capture the correct ground state physics in the ordered phase. However, the performance can be systematically improved by adding instances of the couplings in the ordered phase in the training set (see the Figure below).

We have added the Section “Out of distribution generalization” in the Appendix, where we address the question of whether the model can generalize to points outside the support of the coupling distribution $\mathcal{P}(\gamma)$ used during training. Our results show that such generalization is possible, provided that the training points are sufficiently representative of the test points (for example, in the same phase). This does not extend to cases where the system undergoes a phase transition, as the physical properties change qualitatively across different phases.

However, we again stress that extrapolation outside the support of $\mathcal{P}(\gamma)$ is not expected to be effective in a general setting. A natural and consistent way to increase the applicability of the method, thus, is to extend the support of $\mathcal{P}(\gamma)$ in a way that instances of diverse Hamiltonians (and phases of matter) are considered at training time.

Comment 3 — Inputting Hamiltonian Parameters: The current strategies handle $O(1)$ and $O(N)$ couplings. However, Hamiltonians can have more complex structures (e.g., non-local interactions, varying connectivity) not easily captured by simple vector concatenation or patch-wise splitting. A discussion on limitations and potential alternative representations would strengthen the paper. For example, for a general parameterized Hamiltonian represented as a graph, is there a suitable multimodal architecture to handle such input? (similar scenario occurs for LLM where images can be fed into the model together with texts)

Reply: Indeed the most general two-body Hamiltonian including spatially non-local interactions would require defining $O(N^2)$ couplings. We have not explored this scenario, focusing instead on the important, yet intrinsically more restrictive setting of short-range interactions. As also discussed in the reply to Referee 1, in this work, we set up the key conceptual and technical aspects that, we believe, are important ingredients to develop FNQS. To further extend the framework to wider classes of Hamiltonians, it will be crucial to significantly extend the

training set, and also analyze the optimal manner of encoding the interactions. This is now explicitly stated and discussed in the Conclusions. One possible avenue could involve adapting Graph Transformer Neural Networks (see https://proceedings.neurips.cc/paper_files/paper/2019/file/9d63484abb477c97640154d40595a3bb-Paper.pdf) to handle multimodal inputs. We added a sentence in the work to clarify this aspect.

Comment 4 — Related Work: The authors mention previous attempts [26-29]. It would be beneficial to slightly expand on why FNQS overcomes the limitations mentioned. Furthermore, while distinct from NQS, there is previous work on variational quantum states that shares the same foundation model philosophy of this manuscript, where optimized parameterized quantum circuits/ground state wavefunctions conditioned on problem Hamiltonian parameters are learned by a neural network (<https://journals.aps.org/prapplied/abstract/10.1103/PhysRevApplied.21.014053>). Namely, in the current manuscript, a ground state representation in the form of NQS conditioned on Hamiltonian parameters is learned, while in the previous work, a ground state representation in the form of PQC conditioned on Hamiltonian parameters is learned. The authors should cite this to place the current manuscript in a broader context.

Reply: We extended the discussion about the comparison between our work and the previous attempts. We also included the citation to the suggested work.

Comment 5 — Cost Analysis: Consider adding a brief discussion on the computational cost scaling: How does training an FNQS for R systems with total batch size M compare to training R independent NQS models each with batch size M/R ?

Reply: In general, to achieve stable Variation Monte Carlo optimization, a number of samples on the order of $M \sim 10^4$ is required. Within the framework we propose, a single optimization with $M \sim 10^4$ samples suffices to simultaneously train across R systems, each using only M/R samples. Remarkably, the different systems “help” each other during training, making it possible to achieve stable and accurate optimization even with as few (e.g., $M/R \sim 5$) samples per system (see, for instance, the inset in Fig. 2a). Such a low sample count would not be sufficient to successfully optimize a single system.

Response to Referee 3

Discussion — The authors of this manuscript modify the well-established neural-network quantum states approach by using foundation models and network architectures. With this modification, the authors are able to consider spin configurations and Hamiltonian physical couplings when finding quantum state representations with the foundation model. This way, the approach can generalize to represent quantum states for previously unseen Hamiltonian parameters. They further show that with this approach, the fidelity susceptibility can be evaluated, which can be used to discover phase transitions without including prior knowledge.

With their work, the authors address a rapidly evolving field and provide significant advances in studying quantum many-body systems with the potential for applications in different systems. The proposed generalizability of this method suggests impactful advances in modeling phase diagrams of not well understood systems, which is of significant relevance to various disciplines of physics and beyond. While the claims made in the paper are groundbreaking and the results are legit, the authors need to address the following points in a revised manuscript before it can be considered for publication in Nature Communications.

We thank Referee 3 for their reply. We hereby address in detail the points raised in their report.

Comment 1 — In the abstract, the authors claim that “FNQS can generalize to physical Hamiltonians beyond those encountered during training, offering a unified framework adaptable to various quantum systems and tasks”. However, in the results section, the authors only prove that the method generalizes to parameter choices not encountered during training, but using the same Hamiltonian as during the training. I do not see justification for this claim in the work and the authors should either reformulate this statement or show evidence that the introduced method does generalize to different Hamiltonians. In the manuscript, the authors use similar expressions multiple times (family of Hamiltonians, ensemble of Hamiltonians,...), all these expressions should be addressed.

Reply: This work presents unprecedented applications of a single wave function to a “family” of Hamiltonians. Our meaning of “family” is closely aligned to the mathematical meaning of parameterized objects (see for example the use of the word family in the context of parameterized unitary groups https://en.wikipedia.org/wiki/Stone%27s_theorem_on_one-parameter_unitary_groups). In this sense, we already show full evidence that our approach generalizes to different Hamiltonians, in the sense that Hamiltonians within the same functional form, but with different parameters, are actually different and non-trivially related. As also specified in an answer to Referee 2, we would like to clarify that our approach, like any other Machine Learning approach, is intrinsically limited to instances that are represented by the probability distribution underlying a given training set of problems. It would be inconceivable to use a foundation language model trained, say, on English, Spanish, and Chinese for a downstream task in which the prompts are provided in, say, Hungarian.

We have added the Section “Out of distribution generalization” in the Appendix, where we address the question of whether the model can generalize to points outside the support of the coupling distribution $\mathcal{P}(\gamma)$ used during training. Our results show that such generalization is possible, provided that the training points are sufficiently representative of the test points (for example, in the same phase). This does not extend to cases where the system undergoes a phase transition, as the physical properties change qualitatively across different phases.

However, we again stress that extrapolation outside the support of $\mathcal{P}(\gamma)$ is not expected to be effective in a general setting. A natural and consistent way to increase the applicability of the method, thus, is to extend the support of $\mathcal{P}(\gamma)$ in a way that instances of diverse Hamiltonians (and phases of matter) are considered at training time.

Comment 2 — I don't understand what the most right panel in Figure 1 is showing us. I also don't see it explained in the figure caption. The authors should comment on this.

Reply: With the rightmost panel in Fig.1, our goal is to convey that FNQS, combined with the public availability of pre-trained models such as those we have uploaded to HuggingFace, aims to make quantum many-body physics simulations using neural networks more accessible to the broader physics community. We clarify the role of that panel in the caption of Fig.1.

Comment 3 — The methods section requires some more information. Specifically: - “Consider a family of Hamiltonians, denoted by H_γ ”: What defines a family of Hamiltonians? How are they related? Looking at the results section, they seem to be the same Hamiltonian with different parameter choices. In this case, I would find the expression “family” confusing.

Reply: See the response to **Comment 1**. We believe that the use of family is rather standard in this context.

Comment 4 — The authors say that their goal is to find an approximation of the ground state of an ensemble of Hamiltonians using a variational wave function which explicitly depends on the physical couplings and on a unique set of variational parameters for all the Hamiltonians. Why is this set of variational parameters unique? How is this guaranteed?

Reply: The ansatz we use is by construction featuring a single set of variational parameters that does not depend on the couplings of the Hamiltonian. As in all variational applications, there is no guarantee that the solution found is unique, in the sense that the parameterization is typically redundant and that there might be multiple choices of parameters that correspond to the same physical state. This is by no means a limitation of VMC or any other variational approach. To avoid confusion, we removed the word “unique” from that sentence.

Comment 5 — How is the probability distribution $\mathcal{P}(\gamma)$ over the Hamiltonian couplings defined? How can it be determined in a specific case?

Reply: The formulation we present is intentionally general, and by design, the choice of $\mathcal{P}(\gamma)$ must be specified prior to optimization. Crucially, our method imposes no constraints on the form of $\mathcal{P}(\gamma)$; it can be freely chosen based on the specific context. As discussed in the manuscript, a suitable choice of $\mathcal{P}(\gamma)$ is problem-dependent. For instance, in disordered systems, it typically corresponds to a distribution over the disordered Hamiltonian parameters. In other settings, such as for families of Hamiltonians parametrized by two variables, other forms of $\mathcal{P}(\gamma)$ may be more appropriate, as we illustrate in the relevant examples.

Comment 6 — What are the coupling states? How are they defined? - The coupling states satisfy $\langle \gamma | \gamma' \rangle = \delta(\gamma - \gamma')$, but it is said that γ can be discrete or continuous. In the second last paragraph before II.A, $\mathcal{P}(\gamma)$ is even defined with a sum over γ_k , showing that they are discrete. For the discrete case, the usage of the Dirac delta function $\delta(\gamma - \gamma')$ does not make

sense and should be replaced by the Kronecker $\delta_{\gamma,\gamma'}$.

Reply: We thank the referee for their thoughtful question. In order to avoid any possible source of confusion with the coupling states, which appeared in the first version of our manuscript, we have found a way to reframe the framework to avoid introducing these states. Instead, we now make use of the concept of an ensemble average. Accordingly, we have revised the Methods section of the main manuscript.

For the discrete case, the probability distribution $\mathcal{P}(\gamma) = \frac{1}{\mathcal{R}} \sum_{k=1}^{\mathcal{R}} \delta(\gamma - \gamma_k)$ is defined on continuous couplings, so it is still correct to consider the Dirac delta function. Indeed, when inserted into the loss integral in Eq. (1), this expression yields

$$\mathcal{L}(\theta) = \int d\gamma \left[\frac{1}{\mathcal{R}} \sum_{k=1}^{\mathcal{R}} \delta(\gamma - \gamma_k) \right] \langle \hat{H}_\gamma \rangle_\gamma = \frac{1}{\mathcal{R}} \sum_{k=1}^{\mathcal{R}} \langle \hat{H}_{\gamma_k} \rangle_{\gamma_k}, \quad (1)$$

Comment 7 — In the second last line of the left column on page 3, the authors define the diagonal operator in the computational basis of the system characterized by couplings γ . But the definition is given for $O_\alpha(\sigma, \gamma)$ and in eq. (5) the operator $\hat{O}_{\alpha,\gamma}$ appears. Are those the same operators? Where did the dependence on σ go? Or how does one relate to the other? The same expressions appear around eq. (9).

Reply: The operator $\hat{O}_{\alpha,\gamma}$ is a diagonal operator in the computational basis. Specifically, it is defined through its matrix elements as

$$\langle \sigma | \hat{O}_{\alpha,\gamma} | \sigma' \rangle = \frac{\partial \text{Log} \psi_\theta(\sigma | \gamma)}{\partial \theta_\alpha} \delta_{\sigma,\sigma'}. \quad (2)$$

Notice that the derivatives are with respect to the variational parameters θ . Instead, the operators $\hat{O}_{\alpha,\gamma}$ in Eq. (9) are defined as

$$\langle \sigma | \hat{O}_{\alpha,\gamma} | \sigma' \rangle = \frac{\partial \text{Log} \psi_\theta(\sigma | \gamma)}{\partial \gamma^{(i)}} \delta_{\sigma,\sigma'}. \quad (3)$$

where the derivatives are with respect to the couplings of the Hamiltonian γ . We reported the explicit definition of the matrix elements of these operators in the main text.

Comment 8 — The axis ticks and labels, as well as the legends of the insets are way too small in Figs 2 and 5. They are unreadable in a printed version and should be increased.

Reply: We increased the size of ticks and labels in Figs. 2 and 5.

Comment 9 — In the presented results, the models are always trained closely around phase transitions, including training points in all different phases. While it is great to see that the model then generalizes well, it does not provide evidence for application in broader systems. How does the generalizability behave if the model is only trained on one side of the phase transition? What about training only on points far away from the transition, deep in the

phases? Thinking about general physical systems, where the phase diagram is not known, I would assume that the training points can lie somewhere in the phase diagram and they are not necessarily close to a transition and do not necessarily cover all existing phases. A study on more general choices of training points should be added to this work to make it more consistent.

Reply: We thank the reviewer for this comment, which touches the core of our contribution. We wish to explain that our method, similar to any Machine Learning approach, is fundamentally restricted to cases that are represented in the training set. As also discussed in the reply to Referee 2, it would be inconceivable to use a foundation language model trained, say, on English, Spanish, and Chinese for a downstream task in which the prompts are provided in, say, Hungarian. Still, foundation models are useful because they are able to achieve good predictive performances on several data types (English, Spanish, and Chinese for language models, different phases, and different values of the coupling parameters for our model). However, following the reviewer's suggestion, in the revised version, we added some tests aimed at verifying up to which extent the model can extrapolate to unseen regions. We have addressed it in detail in the Section "Out of distribution generalization" of the Appendix. See also **Comment 2** of Reviewer 2.

Comment 10 — On page 5, end of right column, the authors say that the generated fidelity susceptibility data "can be used to study the collapse of the curves, enabling the extraction of the critical exponents of the Ising model." Have the authors studied this collapse and shown that the expected exponents can be extracted? If such a statement is made, I would expect to see the results to support the validity of the produced data.

Reply: In the revised manuscript, we have added an analysis of the data collapse, demonstrating how it enables the extraction of the critical exponents of the Ising model.

Comment 11 — In Fig. 3, panel a, the color bar is clipped to $[0, 0.5]$. Why is this clipping applied? What would be the plot without clipping?

Reply: Clipping is applied solely for visualization purposes, as the wave function variation associated with the first-order phase transition from columnar to valence bond order is several orders of magnitude larger than the rest of the data. We have verified that this clipping does not affect the results. As an example, in the attached plot of the fidelity for the J_1 - J_2 - J_3 Heisenberg model without clipping we represent the variation intensity on a logarithmic scale instead of applying clipping. A corresponding clarification has been added to the main text.

Comment 12 — How does this method scale to larger system sizes? The authors study up to 10×10 qubits, but do not comment on the computational costs. How expensive are the shown simulations and what kind of scaling is expected when going to larger systems? Especially when studying phase diagrams, which the authors propose as a possible use case of their method due to the generalizability, it is important to study the scaling behaviour to large system sizes. It would thus be essential to know if this method can be efficiently applied to large-scale systems.

Reply: We have added a Section “Systematic energy improvement” in the Appendix, where we study the scaling behaviour of the V-score, a metric that correlates with the accuracy of the wave function. We study the size scaling of the J_1 - J_2 Heisenberg model on the square lattice with periodic boundary conditions from 6×6 up to 12×12 . We observe that the V-score systematically improves when increasing the number of parameters in the network, and only marginally deteriorates with increasing the system size. Regarding the computational cost, we remark that for a fixed neural-network architecture and a fixed total number of Monte Carlo samples M , the computational cost of each simulation remains independent of the number of systems \mathcal{R} . The computational complexity of the ViT scales quadratically with the length of the input sequence of patches. However, several strategies can be employed to mitigate this cost; see Appendix D of Phys. Rev. B 111, 134411 (2025) for a detailed discussion.

Comment 13 — In Fig. 4, right plot of panel a, the x-axis starts counting the seeds at 10. Why is this the case?

Reply: The integer numbers (seeds) shown on the x-axis of the Fig. 4a label different disorder realizations. Specifically, integers from 0 to 6 correspond to a subset of the training disorder realizations, while those from 10 to 16 correspond to a set of test realizations. We have clarified this point in the corresponding caption.

Comment 14 — After eq. (14), the authors set $J = 1/e$. What is e and why is this specific choice made?

Reply: The Random Transverse Field Ising model exhibits a quantum phase transition when $\Delta_J = \Delta_h$, where $\Delta_J = [\ln(J)]_{\text{dis}}$ and $\Delta_h = [\ln(h)]_{\text{dis}}$, with $[\dots]_{\text{dis}}$ denoting the average over disorder [see, Phys. Rev. B 53, 8486 (1996)]. In our work, we choose to introduce disorder solely in the on-site transverse fields h , while keeping the exchange coupling J constant. Specifically, we draw the random fields from a uniform distribution $U[0, h_0]$. If we want to set the critical value $h_c = 1$ we need to choose $J = 1/e$ where $e = 2.718\dots$ is Neper's number.

We have carefully addressed the editorial requests from Nature Communications as well as the remaining comments from the reviewers. In doing so, we have reorganized the manuscript to align with the journal's criteria and have submitted the revised version through the online system. We report a point-by-point response to the latest reviewer comments:

#1: We thank the Referee for their positive assessment and for considering our paper now suitable for publication in Nature Communication. In the resubmitted version of the manuscript, we have addressed the last minor comments by including the limitations of the out-of-distribution generalization using the TFIM model in the Supplemental Material. We also have addressed the issues with the figures.

#2: We thank the Referee for recommending the manuscript for publication in Nature Communications.

#3: We thank the Referee for their response. However, we respectfully disagree with the final comment regarding the generalization capabilities of our method and the interpretation of our previous reply. We believe there may have been a misunderstanding. In our response and in the revised manuscript, particularly in the newly added section "Out-of-distribution generalization", we explicitly demonstrate through numerical experiments that our FNQS can generalize across different Hamiltonians, provided that certain physically motivated conditions are met. We have been transparent in discussing the limitations of this generalization, specifically noting that performance may degrade when the training set does not sufficiently represent the physical properties of the target Hamiltonians. This is a well-known and widely accepted limitation in machine learning and is not specific to our approach. Regarding the related works cited by the Referee (arXiv:2405.21052 and Phys. Rev. B 107, 075147), we have discussed their scope in the Introduction. These studies are confined to relatively simple systems: most notably, the PRB paper focuses exclusively on the 1D transverse-field Ising model and achieve only limited accuracy compared to specialized approaches. In contrast, our work addresses a broader range of systems, including disordered models with N distinct couplings and the highly nontrivial J_1 - J_2 - J_3 Heisenberg model on a two-dimensional square lattice. The latter presents a challenging phase diagram with multiple competing orders, which we are able to capture using a single FNQS architecture. We believe our results extend significantly beyond existing work by offering a systematic investigation of generalization across different Hamiltonians, along with a careful analysis of when this generalization is expected to succeed or fail. We hope this clarifies our contribution.